# Pandemic *Vibrio cholerae* shuts down site-specific recombination to retain an interbacterial defence mechanism

Francis J. Santoriello [1,2], Lina Michel[1,5], Daniel Unterweger[3,4] & Stefan Pukatzki [1,2✉]

*Vibrio cholerae* is an aquatic microbe that can be divided into three subtypes: harmless environmental strains, localised pathogenic strains, and pandemic strains causing global cholera outbreaks. Each type has a contact-dependent type VI secretion system (T6SS) that kills neighbouring competitors by translocating unique toxic effector proteins. Pandemic isolates possess identical effectors, indicating that T6SS effectors may affect pandemicity. Here, we show that one of the T6SS gene clusters (Aux3) exists in two states: a mobile, prophage-like element in a small subset of environmental strains, and a truncated Aux3 unique to and conserved in pandemic isolates. Environmental Aux3 can be readily excised from and integrated into the genome via site-specific recombination, whereas pandemic Aux3 recombination is reduced. Our data suggest that environmental Aux3 acquisition conferred increased competitive fitness to pre-pandemic *V. cholerae*, leading to grounding of the element in the chromosome and propagation throughout the pandemic clade.

[1] Department of Immunology and Microbiology, University of Colorado Denver Anschutz Medical Campus, 13001 E 17th Pl, Aurora, CO 80045, USA. [2] Department of Biology, The City College of New York, 160 Convent Ave, New York, NY 10031, USA. [3] Institute for Experimental Medicine, Kiel University, Michaelisstraße 5, 24105 Kiel, Germany. [4] Max Planck Institute for Evolutionary Biology, August-Thienemann-Straße 2, 24306 Plön, Germany. [5] Present address: Heidelberg University, Grabengasse 1, 69117 Heidelberg, Germany. ✉email: spukatzki@ccny.cuny.edu

**V**ibrio cholerae, the causative agent of the diarrheal disease cholera, causes natural pandemics. Strains of the O1 Classical biotype caused the first six pandemics, and the O1 El Tor biotype currently causes the 7th pandemic[1–3]. Pandemic strains cause diarrheal disease with the virulence factors cholera toxin (CT) and toxin co-regulated pilus (TCP)[4–7]. Several non-O1 strains, however, carry these main virulence factors and cause isolated cases of cholera-like illness without causing pandemic outbreaks[8–10]. The full set of factors driving V. cholerae pandemicity is unknown.

In its aquatic reservoir and the human small intestine, V. cholerae competes with other bacteria and predatory eukaryotic cells via the type VI secretion system (T6SS), a contractile nanomachine resembling a T4 bacteriophage that kills competitors through the contact-dependent translocation of toxic effectors[11–15]. The components of the T6SS are encoded in three clusters (the large cluster, auxiliary cluster 1 (Aux1) and auxiliary cluster 2 (Aux2)), each terminating in an effector/immunity (E/I) pair[16–18]. While T6 effectors are toxic to distinct bacteria, kin cells are protected by cognate immunity proteins[18–20]. It is hypothesised that this allows a strain to propagate clonally[21]. Comparative genomic studies of V. cholerae T6SS clusters show that all pandemic strains carry an identical set of effector genes (A-type), but environmental strains encode variable E/I subtypes[22,23]. Pan-genome phylogeny of V. cholerae does not reflect the dispersion of these effector subtypes[23], suggesting E/I evolution by horizontal gene transfer (HGT). V. cholerae in both the estuarine environment and its human host is exposed to exogenous DNA, bacteriophage and conjugative elements. Further, when in contact with chitin, V. cholerae upregulates the T6SS and natural competence machinery[24–26], driving rapid evolution via inter- and intra-species competition and the uptake of prey DNA. Recently, chitin-induced horizontal transfer of V. cholerae T6SS effectors was demonstrated in vitro[27]. These studies indicate the aquatic environment as a reservoir for the acquisition of new E/I subtypes.

Some T6SS components are bacteriophage structural homologues[12–14], suggesting that the T6SS is the repurposing of one or more prophages. V. cholerae T6SS clusters do not, however, reflect typical prophage genomic organisation or encode functional recombinases. Seventh pandemic El Tor biotype strains encode several genomic islands that do encode phage-like recombination machinery and catalyse site-specific recombination: CTX phage, the SXT element, VPI-1, VPI-2, VSP-I and VSP-II[28–33]. For three of these elements (VPI-1, VPI-2 and VSP-II), integration into and excision from the host chromosome is catalysed by the tyrosine recombinases IntV1, IntV2 and IntV3, respectively[30,31,34]. Tyrosine recombinases do not effectively catalyse excision from the chromosome on their own and require assistance from small DNA-binding proteins called recombination directionality factors (RDFs)[35–38]. Pandemic O1 El Tor V. cholerae strains encode three RDFs (vefA and vefB on VPI-2 as well as vefC on VSP-II), all three of which can promote the excision of both VPI-1 and VPI-2[39,40]. These data support the idea that PAIs encoding bacteriophage-like recombination machinery play an integral role in the development of pandemic V. cholerae strains.

Recently, Altindis et al.[41] identified a fourth T6SS cluster in V. cholerae. This cluster, designated auxiliary cluster 3 (Aux3), encodes a proline–alanine–alanine–arginine motif adaptor protein (PAAR2) that serves to sharpen the tail-spike of the T6SS, a hydrolase (TseH), and its cognate immunity protein (TsiH). TseH is loaded onto the tip of the T6SS with the assistance of the PAAR adaptor and translocated into the target cell, where it catalyses peptidoglycan degradation[42]. Periplasmic localisation of TsiH neutralises this degradation[41]. Unlike the three core T6SS

clusters, Aux3 is not conserved in all sequenced V. cholerae strains[23,43].

Here, we demonstrate that Aux3 extends upstream to include an integrase and a transposase, and that phage-like att sites flank the region from the integrase to tsiH. By analysing 749 V. cholerae genomes, we find that the Aux3 element is encoded in 572 strains of which 566 (99%) encode CT, TCP and a pandemic A-type T6SS effector set[22,23]. Based on phylogenetic analysis of a subset of strains, we show that Aux3 appears to have expanded within the entire pandemic lineage. We further determine that Aux3 is present in nine non-pandemic environmental isolates. The environmental Aux3, however, encodes 42–47 extra bacteriophage homologues and appears to move by HGT, indicating that the pandemic Aux3 is likely the evolutionary remnant of a prophage-like element circulating in the aquatic reservoir. We show that the environmental Aux3 module is excised from and integrated into the host genome by Aux3 integrase-catalysed site-specific recombination. Finally, we show that Aux3 excision in pandemic V. cholerae strains is significantly reduced due to both the loss of an RDF gene and decreased functionality of the pandemic Aux3 integrase. These findings highlight Aux3 as a mobile genetic element (MGE) that was locked into the pandemic V. cholerae lineage making the pandemic form of Aux3 the only T6SS cluster unique to pandemic strains.

## Results

**Phage-like att sites flank T6SS cluster Aux3.** Analysis of Aux3 in O1 El Tor strain N16961 revealed that the genes encoding PAAR2, TseH and TsiH (VCA0284-VCA0286) are immediately downstream from two genes annotated as "phage integrase" (VCA0281, int) and "IS5 transposase" (VCA0282, insH; Fig. 1a). Sliding-window analysis of the region from VCA0280-VCA0287 reveals blocks of variable GC content within Aux3 compared to the surrounding genomic flanks (Fig. 1a). Based on this proximity to putative recombinases and the differential GC content of this region, we hypothesised that this cluster constitutes a potential MGE. Recombinase-encoding MGEs are often flanked by repeat elements (attachment (att) sites) that serve as the locus of enzyme binding and DNA recombination. Alignment of Aux3-encoding and Aux3-naïve V. cholerae strains reveals a single recombinant site on either side of Aux3 indicative of site-specific recombination (Supplementary Fig. 1a). We thus probed the intergenic sequences between gcvT (VCA0280) and int as well as tsiH and thrS (VCA0287) for repetitive sequences and identified two long, direct repeats (referred to as repeat 1 and repeat 2) separated by ~40 bp on either side of Aux3 (Fig. 1b and Supplementary Fig. 1b). Alignment of the Aux3 upstream and downstream intergenic sequences from O1 El Tor strain N16961 with the intergenic region between gcvT and thrS from the naïve chromosome (encoding single copies of repeat 1 and repeat 2) from environmental strain DL4215 shows strong homology upstream and downstream of repeat 2 (Supplementary Fig. 1c). These results indicate that the sequence between the Aux3-flanking repeat 2 sequences is Aux3-derived, while the sequence outside these sites is derived from the Aux3-naïve genome. We propose that repeat 2 is the relevant att site for Aux3 recombination and have renamed the upstream and downstream repeat 2 sequences attL_Aux3 and attR_Aux3, respectively. Importantly, attL_Aux3 and attR_Aux3 are found flanking Aux3 in all analysed Aux3-encoding V. cholerae strains, and attB_Aux3 exists in a single copy between gcvT and thrS in all Aux3-naïve strains (Fig. 1c). These findings demonstrate that Aux3 extends from VCA0281-VCA0286 and potentially constitutes an MGE capable of excising from the genome by site-specific recombination.

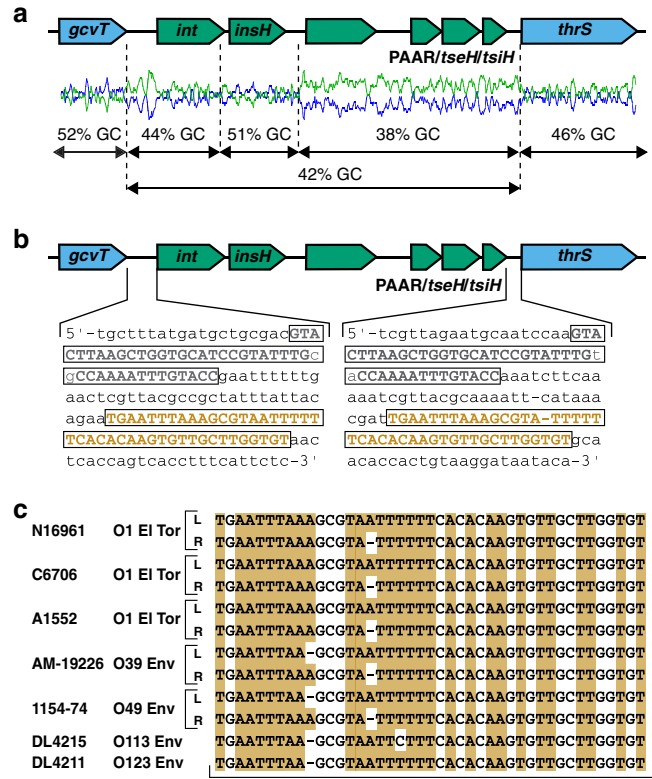

**Fig. 1 T6SS Aux3 module is flanked by conserved phage-like *att* sites.**
**a** Local GC content of the N16961 Aux3 cluster and the flanking regions. Aux3 genes are shown in green and the genomic flanks in blue. GC content (blue line) and AT content (green line) are shown. **b** Intergenic regions flanking Aux3 contain repeated sequences. Direct repeat sequences are boxed and shown in grey (repeat 1) and orange (repeat 2 or *attL_Aux3/ attR_Aux3*). **c** Alignment of phage-like *att* sites from Aux3-encoding strains (N16961, C6706, A1552, AM-19226 and 1154-74) and Aux3-naïve environmental strains (DL4215 and DL4211). Both *attL_Aux3* (top) and *attR_Aux3* (bottom) are represented for Aux3-encoding strains, and *attB_Aux3* is shown for each naïve strain. The average GC content is shown, as *att* sites are typically AT-rich regions.

**Aux3 is conserved and enriched in O1 pandemic strains.** A BLASTN search of the El Tor N16961 Aux3 module in 14 pandemic and 11 environmental *V. cholerae* genomes revealed complete conservation of the Aux3 module across pandemic O1 strains of both the Classical and El Tor biotypes, while all analysed environmental strains lacked Aux3 (Supplementary Fig. 1a). To determine the scale of Aux3 enrichment in pandemic *V. cholerae* strains, we probed the coincidence of *tseH* with the pandemic A-type T6SS effectors *tseL*, *vasX* and *vgrG3*[22,23], as well as *ctxAB* and *tcpA*. We performed a MegaBlast search for these six loci across 749 *V. cholerae* genomes from the PATRIC database[44] to determine the grade (a weighted score accounting for query coverage as well as pairwise identity) for each locus in each genome. Strains were grouped based on having >99% grade to *tseH* as well as >99% grade to the A-type effectors. Of a total 547 strains with hits for *tseH*, 461 strains had a grade of >99% for *tseH*, *tseL*, *vasX* and *vgrG3*, corresponding to an enrichment of *tseH* in pandemic strains of $p = 2.2 \times 10^{-16}$ by Fisher's Exact Test (Supplementary Table 4 and Supplementary Data 1). Due to the fragmented nature of available *V. cholerae* genomes in the PATRIC database, this enrichment is likely an underestimation. We expanded our analysis to include all strains with *tseH* regardless of grade and found that 566 of 572 *tseH*-encoding

strains also encoded *tseL*, *vasX*, *vgrG3*, *ctxAB* and *tcpA* (Supplementary Fig. 2a). It is important to note that the Aux3 element is absent from non-O1/O139 pathogenic strains that do not cause pandemics but carry the major virulence factors CT and TCP (Supplementary Fig. 3). These data demonstrate that Aux3 is enriched in the subset of *V. cholerae* strains with the largest impact on global health.

**Pandemic Aux3 is related to a prophage-like element.** Our MegaBlast search for *tseH*, *tseL*, *vasX* and *vgrG3* in the *V. cholerae* genomes in the PATRIC database revealed six *tseH*-encoding strains that lack *tseL* and *ctxAB* (Supplementary Fig. 2a, b). Three of these strains are environmental O1 strains (2012Env-9, Env390 and 2479-86), two of which encode the toxin co-regulated pilus (2012Env-9 and Env390). The remaining three strains (AM-19226, 1154-74 and P-18748) are non-O1/O139 isolates. Investigation of the region between *gcvT* and *thrS* in these strains revealed an Aux3 cluster ~40 kb in length compared to the 6-kb-long module found in pandemic strains (Fig. 2a). A MegaBlast search for this region in NCBI returned three more strains with this elongated Aux3 element (*V. cholerae* str. 20000, *Vibrio sp.* 2015V-1076, and *Vibrio sp.* 2017V-1038). Importantly, *attL_Aux3* and *attR_Aux3* flank the Aux3 region in each of these strains (Fig. 1c).

Alignment of the Aux3 region in these nine environmental strains reveals variability in the additional sequence between VCA0281 and VCA0284, with most of the variability in the 5′ half of the region (Supplementary Fig. 4a). Further, all environmental strains lack VCA0282 (Supplementary Fig. 4a). Analysis of these nine environmental strains by PHASTER[45] predicts that the Aux3 region in non-pandemic strains resembles an intact prophage of the Myoviridae family (Fig. 2b and Supplementary Fig. 5). Closer examination of the annotated coding regions in the environmental Aux3 elements reveals that the 5′ half of each element is composed primarily of phage regulatory genes like *cro* and *cII*, toxins, methylases, holins and other non-structural genes, but these cassettes vary between strains (Fig. 2c and Supplementary Data 2). The 3′ half of each environmental Aux3 element is more highly conserved and is composed of tailed phage structural genes including capsid, tail, sheath, tube and baseplate (Fig. 2c and Supplementary Data 2). To assess whether this region produces a phage particle, we collected and precipitated supernatants from *V. cholerae* 1154-74 and O395. *V. cholerae* O395 produces the filamentous CTX phage, while 1154-74 encodes a predicted Inovirus (filamentous phage) and the predicted Aux3 Myovirus (tailed phage). We were able to isolate filamentous phage from both O395 and 1154-74, but were not able to detect any tailed phage particles in the 1154-74 supernatant (Supplementary Fig. 4b). Despite its genetic resemblance to an intact prophage sequence, we cannot state that Aux3$^E$ encodes an intact prophage.

We performed a core genome alignment of 69 pandemic and environmental *V. cholerae* strains as well as 8 *Vibrio sp.* and one *V. mimicus* isolate (outgroup), which shows that the incidence of Aux3 in environmental strains is not reflective of phylogeny (Fig. 3). This scenario leads us to conclude that while Aux3$^P$ likely expanded clonally in pandemic strains, Aux3$^E$ may circulate environmentally by HGT. We hypothesise that the evolution of Aux3$^P$ in the pandemic lineage began with the integration of a horizontally transferred phage-like element which then underwent a large deletion event to generate the smaller module (Fig. 2d). The inverse event, in which Aux3$^P$ gained excess prophage-related genes in a large insertion event to form Aux3$^E$, is also a possible scenario. All Aux3$^E$ strains lack *insH* (VCA0282) (Supplementary Fig. 4a), leading us to assume that the insertion

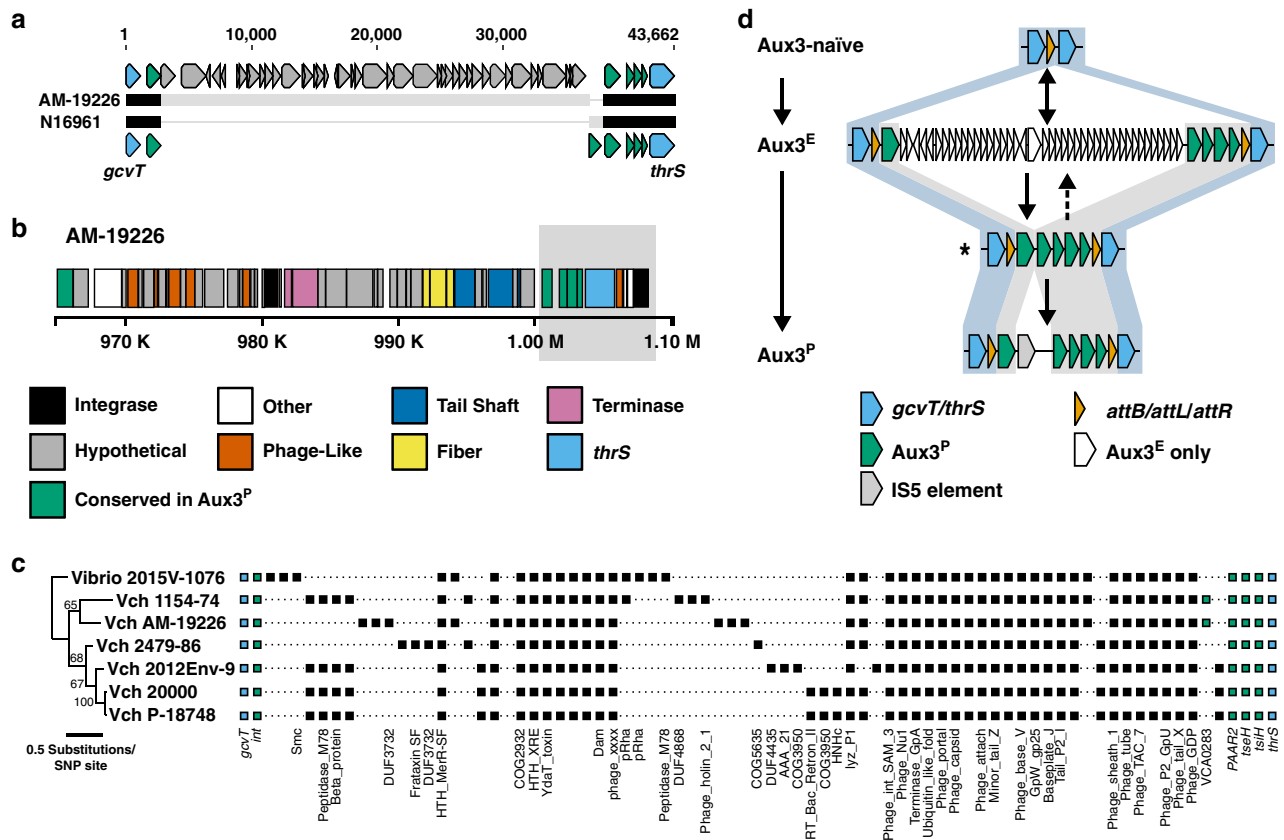

**Fig. 2 Environmental *tseH*-encoding *V. cholerae* strains carry an alternative prophage-like Aux3 element. a** MAUVE alignment of Aux3 from pandemic strain N16961 and environmental strain AM-19226. Flanking genes are shown in blue. Pandemic Aux3 genes are shown in green. Black bars indicate nucleotide agreements and grey bars indicate differences. **b** PHASTER genome diagram showing predicted Aux3 prophage region from the leading integrase (VCA0281) through the superintegron integrase (VCA0291) in AM-19226. Coding regions are coloured according to homology to broad categories of known phage genes. Light grey box indicates region not called by PHASTER but verified manually. **c** Maximum-likelihood tree of Aux3^E modules based on core gene SNPs with associated Aux3^E gene content. Each filled box indicates a gene in the Aux3 module of the associated strain. Conserved genes (>70% amino acid identity) are aligned vertically. Conserved domain hits are indicated. Bootstrapping support values are indicated next to their respective branches. **d** Schematic of proposed Aux3 module evolution from Aux3-naïve environmental to Aux3^P strains. Conserved regions between steps are highlighted in light blue (environmental to pandemic Aux3^P) or grey (Aux3^E to Aux3^P). Asterisk indicates a putative unseen intermediate stage in Aux3 evolution. Dashed arrow indicates alternate hypothesis of a large insertion to form Aux3^E.

of this element occurred in an evolutionary intermediate (Fig. 2d). We have not yet identified a strain encoding this intermediate Aux3^P that lacks the IS5 element. These data support the idea that Aux3 exists in two basic states, environmental Aux3 (Aux3^E) and pandemic Aux3 (Aux3^P), that share a common origin.

**Aux3 is excised from the host chromosome at a defined site.** A BLASTP search for the Aux3 integrase amino acid sequence returned a conserved domain hit for "integrase P4", a common integrase in temperate phages and PAIs known to catalyse integration and excision[30,31,46,47]. During excision, recombination occurs between *attL* and *attR* to reform *attB* at the chromosomal excision junction and *attP* on the excised circular DNA element[48,49] (Fig. 4a). Thus, we aimed to determine if Aux3 excises from the genome to form a circular element. We tested this hypothesis by inverted PCR with primers outside of the *att* sites (P1/P4) and primers inside the *att* sites facing outward (P2/ P3 or P2.2/P3.2; Fig. 4a). With this design, P1/P4 will only be brought into proximity for amplification upon excision and P2/ P3 will only be in the right orientation upon circularisation. We tested two Aux3^E strains (AM-19226 and 1154-74), three Aux3^P strains (N16961, C6706 and A1552), and two Aux3-naïve strains (DL4215 and DL4211)[50] for excision/circularisation. After 4 h of logarithmic growth, excision of the element is detectable in all

Aux3-encoding strains (Fig. 4b). A band indicative of excision is also evident in the tested environmental strains due to the identical nature of the Aux3-naïve and Aux3-excised states. Further, the circular Aux3 module was present in all Aux3-encoding strains and absent from Aux3-naïve strains (Fig. 4b). PCR products were validated by Sanger sequencing against the expected chromosomal and plasmid excision junctions (Supplementary Fig. 6a).

To assess the likelihood of Aux3 module transfer to a naïve strain, we measured the incidence of Aux3 excision in each strain by quantitative PCR (qPCR). Primers were designed against the Aux3-naïve repeats to amplify either repeat 1 or *attB_Aux3* as well as the circular Aux3 junction *attP_Aux3*. This experimental setup allows us to quantify excision (reversion to the naïve state) at each chromosomal site and the presence of circular Aux3 modules (Supplementary Fig. 6b). With two repeat sites in the intergenic flanks, there are two potential integration states of Aux3. Measuring the reversion to a naïve site at both repeat 1 and *attB_Aux3* allows us to confirm the site of recombination. Our results show that when normalised to total genomic DNA, repeat 1 is present at a ratio of ~1 in all tested strains (Fig. 4c), indicating that repeat 1 is constant. The incidence of *attB_Aux3* is ~1/50 genomes for Aux3^E strains and ~1/200 genomes for Aux3^P strains (Fig. 4c), supporting *attB_Aux3* as the site of

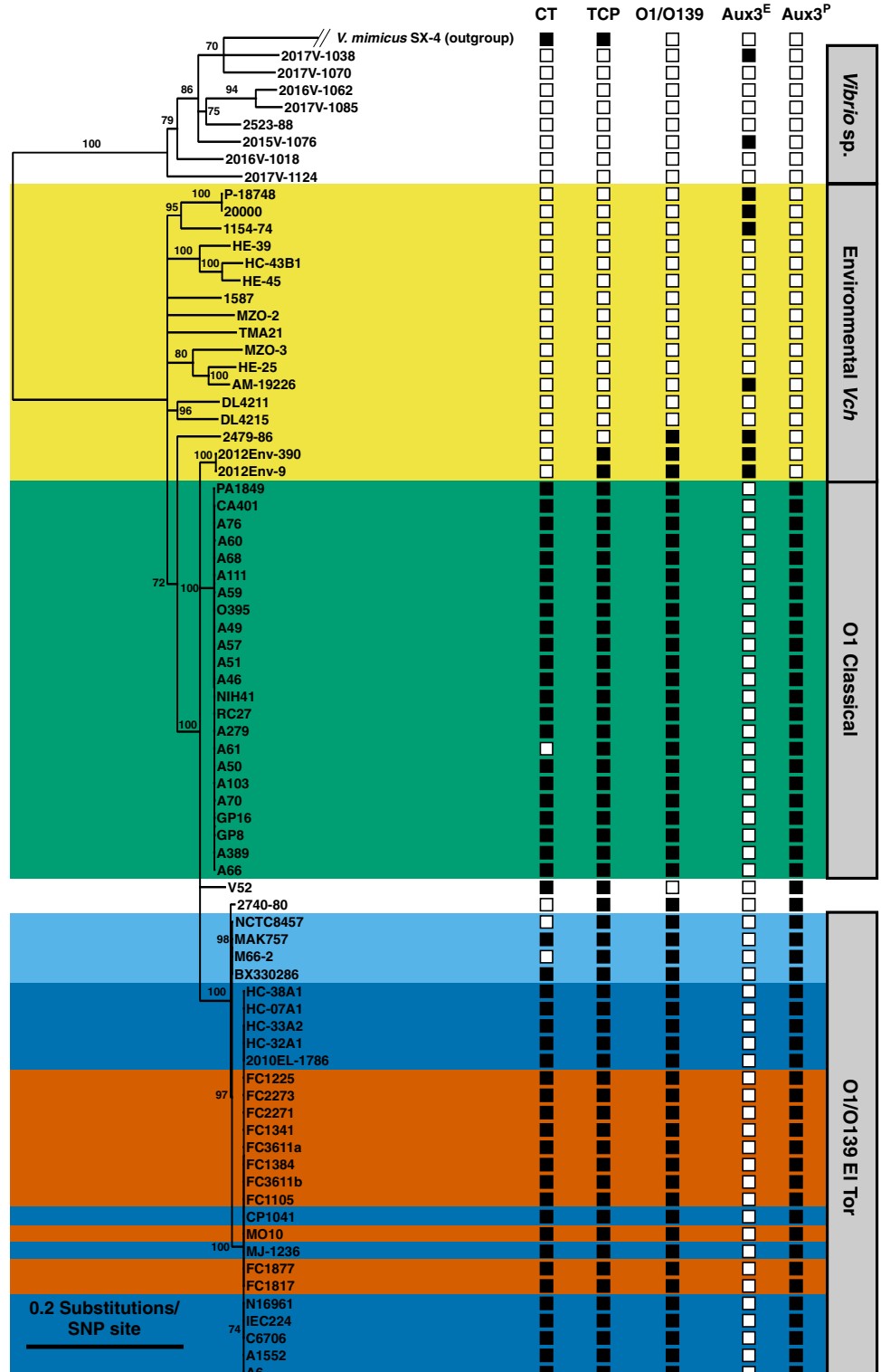

**Fig. 3 The Aux3 element is enriched in pandemic *V. cholerae* and sporadically distributed in environmental strains.** A phylogenetic tree was constructed using the GTR Gamma Maximum likelihood model in RAxML based on core genome SNP alignment of 69 *V. cholerae*, 8 *Vibrio* sp. and 1 *V. mimicus* genome sequences. Bootstrapping support values are indicated next to their respective branches. Nodes with support values <70 were collapsed. Presence (black square) or absence (white square) of CT, TCP, O1/O139 antigen and the Aux3$^E$ or Aux3$^P$ module is indicated. Environmental (yellow), O1 Classical (green), Pre-7th Pandemic O1 El Tor (light blue), 7th Pandemic O1 El Tor (dark blue) and O139 (red) strains are highlighted.

recombination. Time course analysis was performed to assess changes in excision and circularisation in Aux3$^E$ strain AM-19226 during the progression to stationary phase. The portion of genomes with a reformed *attB_Aux3* remains constant over the

AM-19226 growth curve, while the normalised quantity of circular Aux3$^E$ module increases over the AM-19226 growth curve (Fig. 4d). We find that by 4 h of logarithmic growth there is significantly more Aux3$^E$ *attP_Aux3* junctions than there are

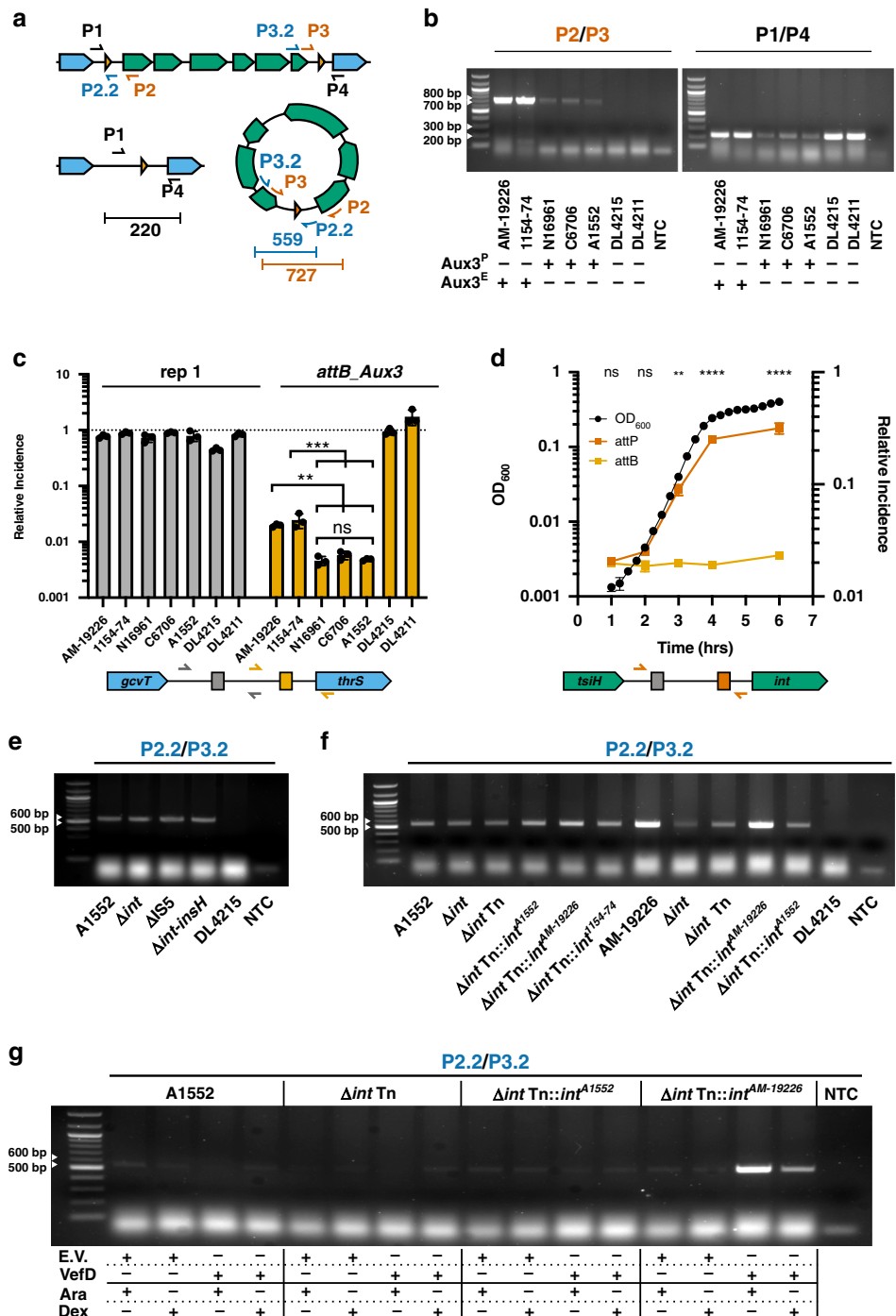

*attB_Aux3* junctions. This may indicate that the Aux3$^E$ module carries an origin of replication, further supporting the idea that Aux3$^E$ is of prophage origin. Finally, while we can detect both the recombined, circular Aux3 module and the chromosomal excision scar in A1552 and AM-19226, we are unable to isolate colonies that have lost the Aux3 module (Supplementary Fig. 6d), leading us to hypothesise that Aux3 is likely excised from the genomes of dying cells.

**Aux3$^E$ and Aux3$^P$ strains catalyse excision differentially.** To investigate the role of the Aux3$^P$-encoded *int* and *insH* recombinases in modular excision, each recombinase was deleted from the A1552 chromosome. Aux3$^P$ circularisation was assessed by inverted PCR with primers over the circular junction. New

circularisation primers (P2.2/P3.2) were designed because the original P2 primer binds within the deleted integrase sequence (Fig. 4a). Neither single recombinase deletion nor a double knockout abolished circularisation of the Aux3$^P$ module in A1552 (Fig. 4e). This could indicate the involvement of an unidentified Aux3-extrinsic recombinase in A1552, as integrase cross-talk between *V. cholerae* PAIs has been previously reported[40]. Deletion of the corresponding *int* gene in the Aux3$^E$ strain AM-19226 largely suppressed modular circularisation, and trans-complementation of the Aux3$^E$ int gene restored circularisation to wild-type levels (Fig. 4f).

These data, along with the excision qPCR (Fig. 4c), suggest that there are disparities in the mechanism of site-specific recombination between Aux3$^P$ and Aux3$^E$ strains. One potential explanation

**Fig. 4 Integrase truncation and the loss of *vefD* leads to reduced excision of Aux3$^P$. a** Inverse PCR schematic showing integrated and excised Aux3$^P$. Aux3 genes are green, genomic flanks are blue and *att* sites are orange triangles. Primers are represented by arrows with expected band sizes below. **b** PCR amplification of excision junctions, *attP_Aux3* (P2/P3) and *attB_Aux3* (P1/P4), on Aux3$^E$ (AM-19226, 1154-74), Aux3$^P$ (N16961, C6706 and A1552) and Aux3-naïve (DL4215, DL4211) strains. **c** Quantification of Aux3 excision by qPCR with primers designed against the naïve repeat 1 (grey) and *attB_Aux3* (orange) (Supplementary Fig. 6b) on gDNA from Aux3$^E$ and Aux3$^P$ strains. Significance was determined by a one-way ANOVA with Tukey's multiple comparisons test (ns non-significant; **$p = 0.0018$, 0.0033 and 0.0020; ***$p = 0.0002$, 0.0003 and 0.0002). **d** Quantification of *attP_Aux3* and *attB_Aux3* in AM-19226 by qPCR in comparison to AM-19226 growth. Growth curve values are shown on the left-axis. Relative incidence of excision values are shown on the right-axis. Schematic of primers designed against the *attP_Aux3* are shown (dark orange) (Supplementary Fig. 6b). Significance was determined by two-way ANOVA with Sidak's multiple comparisons test (ns non-significant; **$p = 0.0042$; ****$p < 0.0001$). **e** Circular excision junction PCR on A1552 wildtype, A1552 single and double recombinase null mutants, and DL4215. **f** Circular excision junction PCR on A1552 and AM-19226 wild-type strains, associated *int*-null mutants, and DL4215. Null mutants from each strain were trans-complemented with an empty mTn7 (Tn), Tn with the native integrase, or Tn with the opposing Aux3-type integrase. **g** Circular excision junction PCR on A1552 and A1552 Δ*int* with trans-complementation of both integrase types, each with and without over-expression of the putative Aux3$^E$ RDF VefD. E.V. = pBAD24, VefD = pBAD24-*vefD*, Ara = 0.1% arabinose, and Dex = 0.1% dextrose. **b, e–g** White arrows indicate ladder band sizes. Gels are representative of at least three distinct experiments ($n = 3$). **c, d** Quantitative results are from three distinct experiments ($n = 3$). Horizontal bars (**c**) or points (**d**) represent the mean and error bars indicate ±SD. **b–g** Source data are provided as a Source Data file.

for this difference is the presence of the IS5 module in Aux3$^P$. A BLASTP search for the *int* amino acid sequence predicts this protein as a P4-like integrase and tyrosine recombinase. Pairwise alignment of the amino acid sequences of pandemic and environmental *int* proteins with other known tyrosine recombinases shows that both have all appropriate catalytic residues intact and strong homology to each other (Supplementary Fig. 7a). At the C-terminus, however, the Aux3$^E$ integrase protein is significantly longer than the Aux3$^P$ homologue. Closer investigation revealed that the IS5 element in Aux3$^P$ inserted immediately downstream of the catalytic Y375 residue, blunting the C-terminal tail of the protein and adding seven nonsense residues encoded by the 5′ end of the IS5 element (Supplementary Fig. 7b). We generated a predictive model of both the full-length and truncated integrase (Supplementary Fig. 7c, d). While the orientation of the catalytic residues is unaffected, IS5 blunting results in a short, disordered C-terminal tail compared to two tyrosine-rich α-helices in the full-length protein (Supplementary Fig. 7c, d), which could explain the decreased incidence of excision seen in Aux3$^P$ strains. To test this hypothesis, we trans-complemented the Aux3$^P$ integrase into AM-19226 Δ*int* and found that it was unable to rescue Aux3 excision, supporting the conclusion that the Aux3$^P$ integrase has lost some functionality (Fig. 4f). In the reverse experiment, trans-complementation of the Aux3$^E$ integrase into A1552 Δ*int* does not appear to raise Aux3$^P$ excision to environmental levels (Fig. 4f). This suggests that the incidence of excision is reliant on both integrase structure and integrase-extrinsic factors that differ between environmental and pandemic strains.

**Loss of an RDF gene contributes to differential excision.** We next aimed to identify the integrase-extrinsic factors that play a role in the reduction of excision between Aux3$^E$ and Aux3$^P$. Aux3$^E$ is much longer than Aux3$^P$ and carries many phage-like genes, and so we hypothesised that Aux3$^E$ may encode a functional RDF gene that was lost in the transition to Aux3$^P$. Loss of the RDF would shift the Aux3 integrase activity towards integration and favour maintenance of the Aux3 module in the chromosome. We first sought to identify a putative RDF gene in the Aux3$^E$ modules. We found that one gene conserved in all 9 Aux3$^E$ modules was predicted to be a helix-turn-helix MerR superfamily protein (Fig. 2c and Supplementary Data 2). The lambda phage RDF (Xis) has been shown to bind DNA via a winged-helix motif of the MerR superfamily[35,38]. We extracted and translated this coding region from all 9 Aux3$^E$ elements and submitted the amino acid sequence to HHpred[51] for further functional prediction. This highly conserved Aux3$^E$ protein

returned three >97% confidence hits for "Regulatory phage protein Cox", "Putative excisionase", and "Phage_AlpA", indicating that this gene may encode an RDF (Supplementary Fig. 7e). We have renamed this gene as Vibrio excision factor D (*vefD*) in agreement with the nomenclature established by the Boyd group. To test our in silico findings, we expressed *vefD* in wild-type A1552 and A1552 Δ*int* with trans-complementation of either an empty mTn7, *int*$^{A1552}$, or *int*$^{AM-19226}$, each expressed from their endogenous promoter (Fig. 4g). We find that co-expression of *vefD* and *int*$^{AM-19226}$ leads to a strong increase in Aux3 excision and circularisation, while all other conditions show wild-type levels of excision for a pandemic *V. cholerae* strain. These results confirm VefD as a functional RDF and further support the reduced functionality of the truncated *int*$^{A1552}$, as co-expression of *int*$^{A1552}$ and VefD did not lead to an increase in excision. These results also show that loss of *vefD* from Aux3$^E$ to Aux3$^P$ was an important step for maintaining Aux3 in the pandemic *V. cholerae* chromosome.

**Aux3 is integrated into an Aux3-naïve chromosome at attB.** To assess the ability of Aux3 to integrate into the chromosome of an Aux3-naïve *V. cholerae* strain, we performed conjugative transfer experiments with an Aux3-null *V. cholerae* recipient strain and a donor *E. coli* S17 λpir carrying a suicide vector with or without an intact *attP_Aux3* site (Supplementary Fig. 8a, b). By conjugating a kanamycin-resistant, circular Aux3 surrogate donor with an Aux3-null recipient, we are able to test the functionality of each integrase type (environmental or pandemic), such that kanamycin-resistant *V. cholerae* clones will only be seen if a trans-complemented integrase can catalyse site-specific, Aux3 integration. To generate a recipient strain, we first replaced Aux3 in *V. cholerae* A1552 with a naïve *attB_Aux3* site from environmental strain DL4211 (A1552 ΔAux3). Next, we introduced a FLAG-tagged copy of either the Aux3$^P$ (A1552) or Aux3$^E$ (AM-19226) integrase back into the chromosome under the control of the P$_{BAD}$ promoter on the mini Tn7 transposon (Tn::*int*$^P$ or Tn::*int*$^E$), allowing us to induce integrase expression with the addition of arabinose to the culture media. Integrase expression was confirmed in these strains by western blot (Supplementary Fig. 8c, d). It is important to note that the Aux3$^P$ integrase construct is expressed at much lower levels than the Aux3$^E$ integrase despite robust expression from the parental plasmid in *E. coli* (Supplementary Fig. 8d). It is possible that the truncated *int*$^P$ is being targeted for degradation in *V. cholerae*. Aux3 donor constructs were generated in pKNOCK-Kan to either carry a stretch of circular Aux3 with *attP_Aux3* intact (pKNOCK-*attP*$^{WT}$) or a deletion of the *attP_Aux3* site

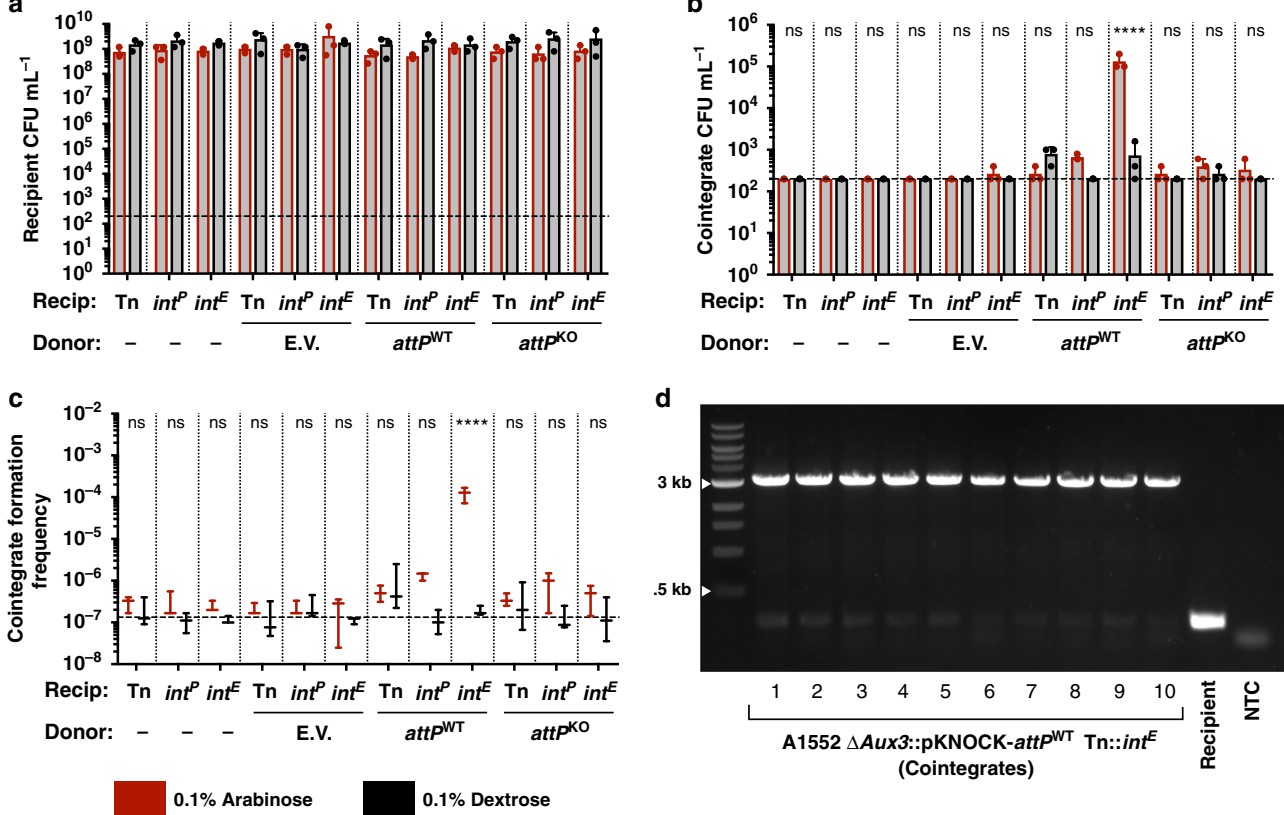

**Fig. 5 The Aux3$^E$ integrase catalyses integration of circular Aux3 into the naïve *attB_Aux3* site. a** Quantification of viable counts of total recipient *V. cholerae* cells (Rif$^R$/Gent$^R$) from conjugative transfer experiments. **b** Quantification of viable counts of cointegrate *V. cholerae* cells (Rif$^R$/Gent$^R$/Kan$^R$) from conjugative transfer experiments. **c** Cointegrate formation frequency from transfer experiments as determined by cointegrate counts divided by total recipient counts. **d** PCR verification of pKNOCK-*attP*$^{WT}$ integration at the defined Aux3 locus with primers P1 and P4 (Fig. 4a). White arrows indicate ladder band sizes. Gel is representative of three distinct experiments (*n* = 3). **a–c** Arabinose-induced experiments are shown in red and dextrose control experiments are shown in black. Horizontal dashed line indicates the limit of detection. Quantitative results are from three distinct experiments (*n* = 3). Horizontal bars represent the mean and error bars indicate ±SD. Significance was determined by a two-way ANOVA with Sidak's multiple comparisons test (ns non-significant, ****$p$ < 0.0001). E.V. = S17 λpir;pKNOCK-Kan; *attP*$^{WT}$ = S17 λpir;pKNOCK-*attP*$^{WT}$; *attP*$^{KO}$ = S17 λpir;pKNOCK-*attP*$^{KO}$; Tn = A1552 ΔAux3 Tn; *int*$^P$ = A1552 ΔAux3 Tn::*int*$^P$; *int*$^E$ = A1552 ΔAux3 Tn::*int*$^E$. **a–d** Source data are provided as a Source Data file.

(pKNOCK-*attP*$^{KO}$; Supplementary Fig. 8b). This experimental design allowed us to determine which integrase can catalyse integration of the Aux3 element into the naïve chromosome and if the recombination happens in a site-specific manner.

After 24 h of co-culture of donor and recipient under inducing or repressing conditions, no significant difference was seen in the recipient or donor counts (Fig. 5a and Supplementary Table 5). While all other conditions resulted in cointegrate formation frequencies at or slightly above the limit of detection, transfer of pKNOCK-*attP*$^{WT}$ to the induced Tn::*int*$^E$ recipient resulted in a 3-log increase of cointegrate formation frequency over baseline (Fig. 5b, c, Supplementary Fig. 8e and Supplementary Table 5). The locus of integration was confirmed by PCR with P1/P4 (Fig. 5d), as an integrated pKNOCK-*attP*$^{WT}$ results in a 3-kb fragment compared to the 220-bp Aux3-naïve fragment. These results demonstrate that the Aux3$^E$ integrase is capable of catalysing recombination between *attP_Aux3* on circular Aux3 and *attB_Aux3* on the naïve chromosome and further indicate that Aux3$^E$ is an MGE circulating in the aquatic *V. cholerae* reservoir.

## Discussion

Here, we demonstrate that the T6SS Aux3 module is largely specific to pandemic strains of *V. cholerae*. We further reveal that this cluster is the evolutionary remnant of a prophage-like element

circulating in the environmental reservoir of non-pandemic *V. cholerae* strains. The Aux3$^E$ element uses its encoded integrase and RDF to catalyse site-specific recombination at the flanking *att_Aux3* sites, forming a circular Aux3 element that is likely primed for horizontal gene transfer to an Aux3-naïve strain of *V. cholerae* (Fig. 6). We show that this cluster is partially conserved and expanded within the pandemic lineage of *V. cholerae*. Despite the lack of the majority of its prophage structural and regulatory genes in the pandemic Aux3$^P$, this cluster maintains a truncated version of its P4-like integrase and flanking *att* sites. Site-specific recombination of this cluster is conserved at lower levels in pandemic strains, although the Aux3 integrase is not necessary for this process (Fig. 6). Finally, we show that the Aux3$^E$ integrase is capable of integrating a circular Aux3 element into an Aux3-naïve chromosome at the *attB_Aux3* site.

The T6SS is a vital defence mechanism for *V. cholerae* and other pathogenic Gram-negative species in both colonisation of the host and interbacterial competition. It is hypothesised that the T6SS is an evolutionary repurposing of a bacteriophage infection[13,14], but the system is conserved so far back in the *Vibrio* lineage that we have not seen evidence of the initial prophage infections that evolved into the system as it exists today. We believe that our findings offer a snapshot of early T6SS evolution, in which a lysogenic phage infection was degraded to solely the components necessary to increase host fitness. Our

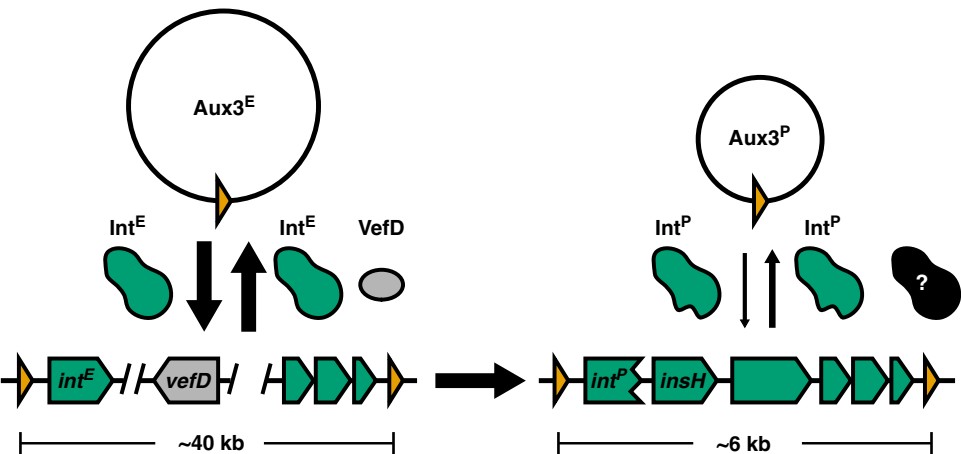

**Fig. 6 Int$^E$/VefD-mediated excision of Aux3$^E$ is shut down in pandemic *V. cholerae* to maintain Aux3$^P$.** Working model of Aux3$^E$ site-specific recombination (left) and degradation of the recombination machinery in the transition to pandemic *V. cholerae* (right). Genes conserved in Aux3$^P$ are shown in green. Aux3$^E$ specific genes are shown in grey. Attachment sites (*att*) are shown in orange. Vertical arrow weight indicates relative quantities of integration and excision. Black protein indicates unknown compensatory recombinases.

results indicate that pandemic Aux3 in *V. cholerae* is related to an environmentally circulating phage-like element that possibly degraded to form the six-gene pandemic-specific module. The route of transfer in the environmental reservoir is currently unknown, as we have no experimental data to support Aux3$^E$ producing its own phage particle. Two potential mechanisms by which Aux3$^E$ could be transferred between strains without making its own phage particle are generalised transduction by environmental lytic phages or chitin-induced natural competence. For the latter mechanism, Aux3$^E$ would be transferred as a linear fragment of genomic DNA and incorporated by homologous recombination in the flanking regions outside of the *attL* and *attR* sites. The *V. cholerae* natural competence machinery can foster the acquisition of linear genome fragments significantly larger than the Aux3$^E$ module[52]. Several *V. cholerae* modules capable of site-specific recombination are transferred by lytic phage transduction[34,53,54], and it is possible that the circular intermediate is more readily packaged into the transducing phage particle.

In *V. cholerae*, a major role for the T6SS is inter-/intraspecies competition and intra-host survival, and the acquisition of new effector proteins could be a key factor in a strain's success or failure in these processes. The phenomenon of T6 effector exchange in *V. cholerae* has been highlighted[22,23,27], but the mechanism has remained elusive. Here we describe a site-specific recombination mechanism of T6SS effector acquisition for Aux3. The acquisition of genomic islands by this mechanism is not uncommon in *V. cholerae*[29-33]. For instance, the GI*Vch*S12 element encodes its own integrase, excises from the chromosome to form a circular element, and carries a cluster of T6SS genes, Aux4, including an *hcp* gene and an E/I pair[32,33,43]. Aux3, however, can be differentiated from GI*Vch*S12 by its distribution. Like Aux3$^E$, GI*Vch*S12 circulates in the environmental reservoir of *V. cholerae* by apparent HGT[23], but Aux3 expands into the pandemic lineage. This indicates that the acquisition of Aux3$^E$ and the eventual reduction to Aux3$^P$ may have been an important step in the transition from an environmental to a pandemic organism.

Further supporting the potential fitness advantage of Aux3, our results show a disparity in the quantity of excision between Aux3$^E$ and Aux3$^P$. Truncation of *int$^E$* appears to have occurred by insertional sequence (IS5 element) interruption to form *int$^P$*. IS5 elements have been shown to drive rapid adaptation in response to environmental stress through either transcriptional regulation of nearby genes or through insertional inactivation[55-58]. Here we

show that the IS5-truncated *int$^P$* is expressed at much lower levels than the full-length *int$^E$*, despite having the same promoter and induction conditions. We speculate that truncation of the *int* gene by IS5 leads to degradation of the Int protein and reduced excision or integration of Aux3. Whether this degradation occurs non-specifically due to truncation and improper folding or as a specific consequence of the short C-terminal tail added by the IS5 element remains to be shown. Integrase-extrinsic factors also appear to be at play in the quantity of excision in pandemic strains. Trans-complementation of *int$^E$* into pandemic *V. cholerae* did not increase excision to Aux3$^E$ levels. Over-expression of *int$^E$* in pandemic *V. cholerae* did, however, catalyse increased integration of our Aux3 surrogate vector. The Aux3 integrase is a tyrosine recombinase, and thus our observation of *int$^E$* only catalysing integration in a pandemic strain background indicates the loss of an RDF[35-40]. We identify the RDF *vefD* in all Aux3$^E$ modules and show that this gene is lost from Aux3$^P$. By co-expressing VefD and Int$^E$ we show that loss of this gene was an important step for shutting down excision in pandemic *V. cholerae*. Our results also indicate that *int$^P$* has reduced functionality, and thus we conclude that Aux3$^P$ excision has been shut down by multiple mechanisms (integrase blunting and loss of *vefD*). We cannot state from the data which of the two events, the loss of Aux3$^E$ genes including VefD or integrase truncation by IS5, occurred first as we have not identified any Aux3$^E$-Aux3$^P$ intermediate modules.

It is likely, based on the two-fold mechanism of Aux3 excision reduction, that genes encoded by the Aux3 element may have conferred a competitive edge to a common ancestor of the pandemic clade. This advantage was locked into the chromosome by IS5 insertion and loss of the RDF *vefD*. This biological phenomenon is referred to as "phage grounding", in which a host cell mutates portions of a lysogen to immobilise advantageous traits on the chromosome[59]. This is one potential route by which the T6SS itself was first acquired. In the case of Aux3, this advantage would likely be conferred by encoding an extra T6SS effector module, as the Aux3 effector TseH was recently shown to effectively kill aquatic competitors from the *Aeromonas* and *Edwardsiella* genus[42]. A new edge against aquatic competitors may have increased abundance or transmission of this pre-pandemic ancestor in the aquatic reservoir. Whether TseH also confers an advantage in human pathogenesis is unknown, but we believe that our findings yield several indications that Aux3 integration was selected for during the evolution of pandemic

*V. cholerae* and that our implementation of further mechanistic studies of the role of TseH in *V. cholerae* transmission and pathogenesis is warranted.

## Methods

**Bacterial strains, plasmids, and growth conditions.** *V. cholerae* strains, *E. coli* strains, plasmids, and primers used in this study are listed in Supplementary Tables 1 and 2. *E. coli* strains DH5α λpir[60], SM10 λpir[61] and S17 λpir[61] were used for cloning and as donor strains in conjugative transfer experiments. All strains were routinely cultured at 37 °C in Lysogeny Broth (LB) with shaking at 250 rpm. Culture on agar plates was done on LB agar at 37 °C or 30 °C. When required, arabinose/dextrose (for the expression or repression, respectively, of *int* under the control of the $P_{BAD}$ promoter) or antibiotics were added to both liquid or agar culture medium at the following concentrations: 0.1% arabinose, 0.1% dextrose, 100 µg mL$^{-1}$ ampicillin, 100 µg mL$^{-1}$ streptomycin, 100 µg mL$^{-1}$ spectinomycin, 50 µg mL$^{-1}$ rifampicin, 50 µg mL$^{-1}$ kanamycin and 10 µg mL$^{-1}$ gentamycin. Instant Ocean (7 g L$^{-1}$) and chitin flakes (8 g per 150 mL, MP Biomedicals) were used for MuGENT cloning experiments.

**Bacterial strain and plasmid construction.** All DNA manipulations were performed according to standard molecular biology protocols. The following enzymes/kits were used according to manufacturer specifications: Phusion High-Fidelity DNA Polymerase (Thermo Fisher), restriction enzymes (New England Biolabs), NEBuilder HiFi DNA Assembly Cloning Kit (New England Biolabs) and Taq PCR Master Mix (Qiagen). Engineered plasmids and bacterial strains were verified by colony PCR and Sanger sequencing (Quintara Biosciences).

Genetic deletions in *V. cholerae* were generated by either allelic exchange and sucrose counter selection with the suicide vector pCVD442[62] or the natural transformation-based MuGENT method[63]. The plasmid pKD13[64] served as template for the amplification of the flippable antibiotic cassette FRT-Kan-FRT. For allelic exchange, the flippable Kan cassette was inserted between 1 kb homology arms in pCVD442 by Gibson cloning[65], and clones were selected by sucrose counterselection and kanamycin resistance. The MuGENT technique was used to generate A1552 ΔIS5 and A1552 ΔAux3. Unmarked ΔIS5 construct was generated by Gibson cloning of 3 kb fragments upstream and downstream of the Aux3 IS5 element into pUC19 and amplification of a 6-kb fragment with pUC19 specific primers. Unmarked ΔAux3 construct was generated by amplifying a 6-kb fragment containing naïve *attB_Aux3* site from the DL4211 chromosome. Selective fragments were generated by Gibson cloning a spectinomycin resistance cassette from SAD033 in between 3 kb homology arms encompassing the *V. cholerae lacZ* gene and the surrounding sequence into pUC19. The 5.8-kb selective *lacZ::Spec* fragment was amplified using primers ABD334/ABD335[63]. Unmarked and selective fragments were co-transformed into *V. cholerae* cells on chitin. White/spectinomycin resistant cells were screened for the unmarked mutation. The spectinomycin resistance cassette was cured from *lacZ* with pCVD442 carrying a wild-type copy of *V. cholerae lacZ*.

All trans-complementation vectors were generated by Gibson cloning. Expression constructs with either endogenous or $P_{BAD}$ promoters were inserted into the mini Tn7 transposon (mTn7/Tn) in pGP704-mTn7. Constructs were moved onto the *V. cholerae* chromosome by tri-parental mating[66–68]. In the case of VefD, the *vefD* gene from AM-19226 was inserted into pBAD24, and this vector was moved into *V. cholerae* by electroporation.

Donor pKNOCK-*attP* vectors were generated by Gibson cloning. The circular Aux3 *attP* region was also generated by Gibson cloning. Primers overlapping the *att* site were modified to remove the *attP_Aux3* site. Regions containing either *attP*$^{WT}$ or *attP*$^{KO}$ sites were amplified off the Gibson assembled fragments and assembled into the SmaI-cut pKNOCK-Kan vector.

**Identification of *att* sites and bacteriophage elements.** To identify potential *att* sites, intergenic sequences from VCA0280 to VCA0281 and VCA0286 to VCA0287 were concatenated and input into REPFIND[69] with a minimum repeat size of 10 and a *P*-value cutoff of 0.0001. To identify putative prophages, GenBank files for Aux3$^E$ strains were submitted to PHASTER[45].

**Nucleotide/amino acid sequence alignment.** All genomes used for alignments can be found in Supplementary Table 3. All nucleotide alignments outside of phylogenetic analyses were performed in Geneious Prime (v2019.0.4). Nucleotide sequences encompassing more than one open reading frame were aligned using the Progressive MAUVE algorithm[70] to account for insertions, deletions and rearrangements. Single gene, intergenic region or single protein sequence pairwise alignments were performed using MUSCLE (v3.8.425)[71].

**Aux3 enrichment analysis.** MegaBlast queries were performed in Geneious Prime (v2019.0.4). Downstream manipulations and plots were done in RStudio (R version 3.3.2 (2016-10-31) -- "Sincere Pumpkin Patch"). *V. cholerae* genomic FASTA files were downloaded from the PATRIC database[44]. Nucleotide sequences for *tseH* (VCA0285), *tseL* (VC1418), *vasX* (VCA0020), *vgrG3* (VCA0123), *tcpA* (VCA0828), and *ctxAB* (VC1456-VC1457) from O1 El Tor type strain N16961 were queried by

MegaBlast against a custom database of PATRIC FASTA sequences to generate a grade (a weighted metric combining query coverage (0.50), e-value (0.25) and pairwise identity (0.25)) for each gene locus in each strain. Strains were grouped based on a 99% grade cutoff for *tseH* and the three A-type effectors *tseL*, *vasX* and *vgrG3* to create 4 groups (Supplementary Table 4) and assess occurrence of *tseH* in AAA pandemic strains by Fisher's exact test. PATRIC strains were k-means clustered by Partitioning Around Medoids (pam, R package cluster v2.1.0) based on grades for *tseH*, *tseL*, *vasX*, *vgrG3*, *ctxAB* and *tcpA*. Mean grade was determined at each locus for each cluster and plotted as a heat map (pheatmap, R package pheatmap v1.0.12).

**Phylogenetic analysis and tree building.** Genomic FASTA files for tree building were obtained from the PATRIC database[44] or NCBI and annotated using Prokka (v1.12)[72]. A core genome was extracted from Prokka-output GFF3 files and aligned using Roary (v3.11.2)[73]. The core genome alignment was reduced to loci harbouring polymorphisms using SNP-sites (v2.4.1)[74]. Phylogenetic tree was built using the RAxML (v 7.0.4) GTR Gamma Maximum Likelihood model. Statistical branch support was obtained from 100 bootstrap repeats. Phylogenetic trees were visualised from RAxML-generated newick files using TreeGraph 2 (v2.15.0-887 beta)[75]. Branches with bootstrapping support values <70 were collapsed. Presence of TCP and CTX were determined by MegaBlast for *tcpA* (VC0828) and *ctxAB* (VC1456-VC1457). O1 antigen status was determined from the literature. Presence of *tseH* was determined as described above.

**Functional prediction of phage genes in Aux3E modules.** Aux3$^E$ genomic regions were extracted from *gcvT* to *thrS* and regions were re-annotated by Prokka (v1.12)[72]. All annotated genes (from both original Genbank files and re-annotated files) were extracted and translated. Amino acid sequences for all extracted annotations were submitted to NCBI Conserved Domain Search (https://www.ncbi.nlm.nih.gov/Structure/cdd/wrpsb.cgi) to identify putative functional domain hits. Further functional prediction of select genes was performed by submission to HHpred[51] (MPI Bioinformatics Toolkit, https://toolkit.tuebingen.mpg.de/tools/hhpred).

**Excision/circularisation PCR and quantitative PCR.** Bacterial strains analysed by excision/circularisation PCR or qPCR were grown overnight as described above. Approximately equivalent growth for all analysed strains was verified (Supplementary Fig. 6c). For Fig. 4b, e, f, g overnight cultures were subcultured 1:50 in 5 mL of fresh LB and grown for 4 h. For Fig. 4g, 0.1% arabinose or 0.1% dextrose was added at the 1-h time point. Cultures were normalised to OD$_{600}$ and pelleted (4300 × g, 10 min) and resuspended at 10X concentration in nuclease free H$_2$O. Cell suspensions were boiled for 5 min to release nucleic acids. PCR was performed on 3 µL of each lysate with the indicated primers. In all, 3% DMSO was added to reactions using P2.2/P3.2 due to lower primer efficiency. For excision/circularisation qPCR, overnight cultures were subcultured 1:50 in 5 mL of fresh LB and grown for 6 h. In all, 1 mL of culture was collected (at 1, 2, 3, 4 and 6 h) and pelleted (14,000 rpm, 2 min). DNA was extracted by phenol/chloroform extraction. All DNA samples were normalised to 20 ng µL$^{-1}$ and 50 ng µL$^{-1}$. qPCR was performed on 250 ng (Fig. 4c) or 100 ng (Fig. 4d) of each sample in a 20-µL reaction volume with Bio-Rad SYBR Green Master Mix according to the product manual. Primers targeted repeat 1, *attB_Aux3*, and *ompW* or *attP_Aux3* and *ompW*. Data was collected in Bio-Rad CFX Manager 3.1. All targets were measured by absolute quantification against the following standard curves: A1552 ΔAux3 genomic DNA (Aux3-naïve) for repeat 1, *attB_Aux3*, and *ompW* and pUC19-*attP* plasmid DNA for *attP_Aux3*. Repeat 1, *attB_Aux3*, and *attP_Aux3* signal was normalised to *ompW* to control for variability in input DNA. Averages of at least three independent experiments (±standard deviation) are provided.

**Aux3 excision tracking by colony-forming unit counts.** Strains with an Aux3 internal kanamycin resistance cassette were struck out for isolated colonies on LB agar plates with the addition of rifampicin and kanamycin (A1552) or streptomycin and kanamycin (AM-19226). Three individual clones were selected for each tested strain. Clones were inoculated in 5 mL of LB media with rifampicin (A1552) or streptomycin (AM-19226) and grown shaking at 37 °C. At 24 hr each culture was subcultured at a ratio of 1:100 into 5 mL of fresh LB with the above indicated antibiotics. Remaining culture was serially diluted at a 1:10 ratio to 10$^{-7}$. Dilution series were spotted (5 uL) on LB agar plates both with (Aux3-maintained colony-forming unit (CFU)) and without (total CFU) the addition of kanamycin. This process was repeated at the 48-h time point. Colonies were counted from the highest countable dilution spot to determine viable CFU.

**Aux3 module transfer experiments.** Overnight cultures of recipient strains (A1552 ΔAux3 with variable mTn7 constructs) and donor strains (S17 λpir with variable pKNOCK vectors) were pelleted (4300 × g, 10 min) and resuspended at 10X concentration in LB media. In total, 10 µL of each concentrated cell suspension was resuspended in 1 mL LB (1:100), from which serial dilutions were prepared and plated as spots on LB agar to determine input CFU. Donor and recipient strains were mixed in all combinations at a ratio of 10:1 donor to recipient. Mixtures were plated as 25 µL spots on Durapore 0.22 µm PVDF filters (Millipore Sigma) on

pre-dried, pre-warmed LB agar plates with either arabinose or dextrose. Spots were dried and incubated at 37 °C for 24 h. After 24-h incubation, filters were collected and resuspended by vortexing in 1 mL LB media. Serial dilutions were prepared from each suspension, and dilution spots were plated on LB agar with kanamycin (donors), rifampicin + gentamycin (recipients), or rifampicin + gentamycin + kanamycin (cointegrates) to determine CFU for each subset of cells. Colonies were counted from the lowest countable dilution. Cointegrate formation frequency was determined by dividing cointegrate CFU/mL by total recipient CFU/mL. Averages of at least three independent experiments (±standard deviation) are provided (Supplementary Table 5).

**Reporting summary**. Further information on research design is available in the Nature Research Reporting Summary linked to this article.

## Data availability

The authors declare that all the data supporting the findings of this study are available within the paper and its supplementary information files. All genomes analysed in this study are publicly available from the PATRIC (https://www.patricbrc.org/) and NCBI RefSeq (https://www.ncbi.nlm.nih.gov/refseq/) databases. Source data are provided with this paper.

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

## Acknowledgements

We acknowledge Michelle Dziejman, David Rozak, Daniele Provenzano and Melanie Blokesch for strains and plasmid constructs necessary for the completion of this study. This work was supported by the National Institutes of Health (R01AI139103) and the University of Colorado Anschutz Medical Campus.

## Author contributions

F.J.S., D.U. and S.P. designed the experiments. F.J.S. and L.M. performed the experiments. F.J.S. and S.P. analysed the data. F.J.S. and S.P. wrote the paper.

## Competing interests

The authors declare no competing interests.
