## [Peer Review File · Nature Communications]

Reviewers' comments:

Reviewer #1 (Remarks to the Author):

In this study, Santoriello et al report an important mechanism of T6SS effector-acquisition in pandemic *Vibrio cholerae* strains which may account for the spreading and competitive fitness of the pandemic *V. cholerae* against environmental isolates. The authors first used genome comparison approaches to determine that the auxiliary cluster 3 (Aux3) is enriched in pandemic strains specifically in a prophage-like region. This region also exists in a small set of environmental strains but absent in the majority of environmental strains. They further demonstrated that Aux3 could be excised from the chromosome of both pandemic and environmental *V. cholerae*, but only the environmental Aux3 could integrate into the Aux3-naïve chromosome. As the T6SS is critical for *V. cholerae* competing against other bacteria and facilitate horizontal gene transfer (HGT) by lysing competitors to gain access to gDNA, there is some limited evidence that suggest T6SS effector modules can be acquired/exchanged through HGT. However, this has not been previously demonstrated experimentally by the field and this study on Aux3 fills this gap. The study is well written and should be interesting to the field. Below are some comments and suggestions to the authors.

1. Line 202, as the authors estimated that excision occurred at a frequency of 1/100 genomes for the environmental strains and 1/500 for pandemic strains. In other words, from a tube of 1 ml culture, there might be 1E7 cells without Aux3. We should then expect more pandemic isolates without this Aux3, unless there is some selection pressure for retaining it in the genome. The authors might elaborate on why pandemic strains all have Aux3 despite of the excision and where the selection pressures could come from. One interesting experiment might be testing excision after extended period of incubation to see if the excised mutant would be enriched.
2. In Fig 5b&C, there is somewhat obvious increase (5-10fold) in the strain expressing pandemic integrase, though the statistic test is non-significant. Considering that the expression level of intP is much lower than intE, it is quite possible that intP is functional but the phenotype is ns due to low expression. However, in the abstract (line 36) and intro (line 91), the authors made strong statement that only the IntE works. In addition, the authors discussed the low level of intP might be due to degradation. If it is just a quantity issue, the authors might also try to express intP on a plasmid vector.
3. In Fig 4f, there is some background difference between A1552 and the AM19226 that inducing the intAM19226 didn't work in the A1552 mutant but complemented in the AM19226 mutant. The authors should test the effect of intA1552 expression in the AM19226 mutant background to confirm that the intA1552 doesn't work or not as efficient as the intAM19226 enzyme.
4. It is not clear how the relative incidence fig4c was calculated. Please provide a little more description. It seems that qPCR quantification was done using a standard curve method. In that case, shouldn't it be that primer efficiency doesn't contribute to underestimation since the same primers were used for the samples and the standard curve?

Minor

1. In Fig 4e, the ins-is4 deletion mutant also showed excision in A1552. Could the authors discuss what other extrinsic enzymes might be responsible for this?
2. Since the authors tried to address the discrepancy between intA1552 and intAM19226 by comparing the sequences and predicted structural models, have the authors considered an in vitro biochemical assay to test the enzymatic activity of intA1552?
3. Line 380, the gene name for TseH is VCA0285. Please correct this.
4. In Fig4d, it is not clear whether total gDNA or ompW was used as control to show that the increasing signal intensity is specific to increased excision but not due to overall increase in DNA quantity.
5. Line 79, this statement is inaccurate. TseH showed toxicity to cells only when expressed in the periplasm but not in cell-to-cell competition in the listed reference.

6. Fig 5d. please provide the size of the DNA marker corresponding to the target band at least.
7. Line 216. Primer P3.2?
8. Fig 2C. Shouldn't the Aux3-naïve att be attC? If so, indicate this in the legend.

Reviewer #2 (Remarks to the Author):

This is a well-written thought provoking paper on the evolution of T6SS and their components. The authors identify T6SS genes that are associated predominantly with pandemic strains and describe the evolutionary history of the region in *V. cholerae*. The paper also describes a mechanism of how these genes became fixed in the population.

A deeper description of the phages carrying the AUX3 would be helpful. How similar are the phages, do they belong to the same viral family. Are their integrases homologous? In strains that contain AUX3 associated with a prophage are there conserved att sites. The location of AUX3 genes in the terminal ends of the prophage suggest they were pick up by imprecise excision. Has the circular intermediate of any of these phages been examined? Do you get AUX3 genes 1-5 forming part of the circular intermediate when the prophage excised.

Line 119 –Is attC2 in the AUX3 naïve strains identical to attL2 in size and sequence? A bp deletion in attC would indicate that a MGE was previously inserted there and when it excised it caused a deletion. If they are both identical, it suggests AUX3 was never present in these strains.

Line 213 Is it possible that deletion of int disrupted the att site? Did you identify an excisionase associated with either MGE? Excisionases/recombination directionality factor are usually require for excision of TR integrases. In addition, another integrase elsewhere in the genome might perform the excision –cross-talk between MGE has been documented in *E. coli* and *V. cholerae* isolates. What is the integrase composition of the strains tested?

Use consistent nomenclature for strain names i.e. AM199226 should be AM-19226

Line 71-72 should include Carpenter, MR et al., 2017

Line 567 Ref. missing journal name

Line 583 ref. missing information

Line 587 re. missing journal title

Reviewer #3 (Remarks to the Author):

NCOMMS-20-02368-T ms describes the discovery that the conserved Aux3 cluster of T6SS effectors is specific of the *V. cholerae* pandemic strains and that it is carried within a mobile genetic element (MGE) remnant of a larger one, possibly a phage, circulating among a few environmental strains. This, per se, is not extremely original, as many “pathogenicity islands” carrying virulence and pathogenicity accessory functions have been described over the last 30 years in many bacterial models - including *cholerae*, with the CTX phage and the TCP locus. That said, this an interesting discovery, especially in the context of the role of T6SS effector sets for the killing of bacteria in complex communities. Both the comparative genomic study and the approaches chosen to identify the boundaries of the MGE, which are fairly classical, are sound and convincing.

Though if the study is well performed, the description of the identification of the recombined sequences is really confusing, and must be clarified. There are also several misuses of the terminology, both for recombination sites and gene names, which gives an impression of amateurism, that need to be corrected.

My comments below may help

Results

L108 and following

Please do not call the repeat sequences 1 and 2 attL/R1 and attL/R2, as you show later in the ms that the only real attL/R is attL/R2. This leave the impression that there are two attL/R. Call these repeats 1 and 2 and then attL/R_Aux3 for instance, once it is shown to be the integration borders and rewrite the whole paragraph accordingly.

L113 L180, ...:

attC: this terminology is already used for integron Cassette recombination sites (see the Mazel group review in the ASM mobile DNA 3). For such MGE or lysogenic phage in general, the proper term is attB (B for Bacteria by opposition to attP for Phage). And again, no need of attB1 and attB2, as there is a single one functional site according to your assay.

L111

about the stretch of T, please delete this useless part. HK022 which is highly related to Lambda (much more than P4) has an overlap region with more G, ie it has no meaning and give the impression that it is somewhat related to lambda...

Fig 1 and text. Please correct the gene names: is5 is not a gene! So, use insH, which is the name of the IS5 transposase (or VCA0282).

L269:

This is not a conjugation frequency, but a cointegrate formation frequency, please correct.

Discussion

L280.

As you have no proof that the MGE is a phage, even in the environmental form (as mentioned L300), just delete phage before integrase.

L303 and 304. Transformation goes through ssDNA, exclusively, there is de-gradation of one of the two DNA strand, so apart if they produce circular concatainers, the chance to transfer the circular form are null.

L331 and following.

You do not discuss the possible presence of a recombination directionality factor encoding gene (ie a xis gene) in the Aux3E locus, which if absent in the P version could explain your results? IntE would not be sufficient for excision in P strain, but sufficient for integration, ie similarly to what you observed. This is worth discussing, and furthermore, more generally you should discuss the genes/ORF found in the Aux3E MGE, what are the chance that it is really a lysogenic phage?

Reference list

Ref 13: "Proc...." ?

Ref 22: the journal (Science) is missing

Ref 28 : the journal (Mol Micro) is missing

Ref 44: the journal (PNAS) is missing

Thank you to all of the reviewers for your helpful comments and suggestions. We have addressed each of your comments as described below, and we believe that the edits resulting from your suggestions have significantly strengthened our findings and conclusions. We sincerely appreciate your time and your contribution to improving the quality of this study.

Reviewers' comments:

Reviewer #1 (Remarks to the Author):

In this study, Santoriello et al report an important mechanism of T6SS effector-acquisition in pandemic *Vibrio cholerae* strains which may account for the spreading and competitive fitness of the pandemic *V. cholerae* against environmental isolates. The authors first used genome comparison approaches to determine that the auxiliary cluster 3 (Aux3) is enriched in pandemic strains specifically in a prophage-like region. This region also exists in a small set of environmental strains but absent in the majority of environmental strains. They further demonstrated that Aux3 could be excised from the chromosome of both pandemic and environmental *V. cholerae*, but only the environmental Aux3 could integrate into the Aux3-naïve chromosome. As the T6SS is critical for *V. cholerae* competing against other bacteria and facilitate horizontal gene transfer (HGT) by lysing competitors to gain access to gDNA, there is some limited evidence that suggest T6SS effector modules can be acquired/exchanged through HGT. However, this has not been previously demonstrated experimentally by the field and this study on Aux3 fills this gap. The study is well written and should be interesting to the field. Below are some comments and suggestions to the authors.

1. Line 202, as the authors estimated that excision occurred at a frequency of 1/100 genomes for the environmental strains and 1/500 for pandemic strains. In other words, from a tube of 1 ml culture, there might be 1E7 cells without Aux3. We should then expect more pandemic isolates without this Aux3, unless there is some selection pressure for retaining it in the genome. The authors might elaborate on why pandemic strains all have Aux3 despite of the excision and where the selection pressures could come from. One interesting experiment might be testing excision after extended period of incubation to see if the excised mutant would be enriched.

Thank you for this observation/suggestion. We too were curious about the selective pressures leading to Aux3-maintenance in the genome. After the submission of our manuscript, Tao Dong's laboratory published a manuscript on the species-specific killing capacity of the Aux3 effector TseH (Hersch *et al.*, 2020). This study shows that TseH, while ineffective at killing *E. coli* and non-immune *V. cholerae*, is very effective at killing *Aeromonas* and *Edwardsiella* species. Both of these species would likely encounter *V. cholerae* in the environmental reservoir. We have added this citation to the text (Line 84, 411).

To test whether long term culture leads to enrichment for Aux3-excised strains, we cultured A1552 Aux3::Kan (Kan cassette inserted between VCA0283 and VCA0284), AM-19226 *int*::Kan, AM-19226 *int*::Kan mTn7, and AM-19226 *int*::Kan mTn7::*int*^{AM-19226} for 48 hr in shaking culture at 37 C with subculture into fresh media at 24 hr. Colony forming units were counted at the 24hr and 48hr time point for total cells (Rif or Strep resistant) and Aux3-maintained cells (Rif/Kan or Strep/Kan). We found that at both 24hr and 48hr there was no significant difference in the total cells vs. Aux3-maintained cells for any of the mutants (Supplementary Fig. 6d). This leads us to conclude that the loss of Aux3 after acquisition is a rare event. It is likely that Aux3 is excised in response to impending cell death and cells that have excised Aux3 do not propagate further. Based on our results we cannot conclude that Aux3 is never lost due to excision, only that we do not detect that loss with our chosen experiments.

Added Line 233: “Finally, while we can detect both the recombined, circular Aux3 module and the chromosomal excision scar in A1552 and AM-19226, we are unable to isolate progeny cells that have lost the Aux3 module (Supplementary Fig. 6d), leading us to hypothesize that Aux3 is likely excised from the genomes of dying cells.”

2. In Fig 5b&C, there is somewhat obvious increase (5-10fold) in the strain expressing pandemic integrase, though the statistic test is non-significant. Considering that the expression level of intP is much lower than intE, it is quite possible that intP is functional but the phenotype is ns due to low expression. However, in the abstract (line 36) and intro (line 91), the authors made strong statement that only the IntE works. In addition, the authors discussed the low level of intP might be due to degradation. If it is just a quantity issue, the authors might also try to express intP on a plasmid vector.

Thank you for your comment on the strength of the claim that only Int^E is functional. We agree that there is an evident 5-10 fold increase in integration in the strain expressing the pandemic integrase. This same increase, however, is seen in the empty transposon control. Fluctuation above the limit of detection is seen in all experiments with a donor vector carrying an *attP* site. We believe our added data showing the inability of Int^P to rescue excision in AM-19226 shows that the integrases activity is severely reduced (See comment 3). Our results lead us to conclude that multiple recombinases may be responsible for integration/excision in pandemic *V. cholerae*. It is possible that this basal level of integration in any strain receiving a donor vector with an intact *attP* site is due to compensation by a different enzyme. To the reviewer's point regarding the use of a plasmid vector, we generated new Int^E and Int^P over-expression vectors (pBAD24-FLAG-*int*^{A1552} and pBAD24-FLAG-*int*^{AM-19226}). The same difference in expression levels was seen between Int^P and Int^E (Supplementary Fig. 8c). Again, we believe that the strongest piece of data about the functionality of Int^P comes from our new data showing that Int^P driven from its native promoter cannot complement the loss of Aux3 excision in an AM-19226 background. We do, however, agree that our statement that only Int^E is functional is too strong based on the presented data. We have edited the text accordingly:

The first instance of this statement was removed from the abstract in the process of editing it down to the appropriate word count.

Line 95:

~~“Finally, both the environmental and pandemic versions of Aux3 excise from the genome by site-specific recombination, but only the environmental Aux3 integrase can catalyse transfer to a naïve strain.”~~

“We show that the environmental Aux3 module is excised from and integrated into the host genome by Aux3 integrase-catalysed site-specific recombination. Finally, we show that Aux3 excision in pandemic *V. cholerae* strains is significantly reduced due to both the loss of an RDF gene and decreased functionality of the pandemic Aux3 integrase.”

3. In Fig 4f, there is some background difference between A1552 and the AM19226 that inducing the intAM19226 didn't work in the A1552 mutant but complemented in the AM19226 mutant. The authors should test the effect of intA1552 expression in the AM19226 mutant background to confirm that the intA1552 doesn't work or not as efficient as the intAM19226 enzyme.

Thank you for your suggestion to complement *int*^{A1552} in the AM-19226 *int* background. We have performed this complementation experiment. Fig. 4f now includes AM-19226 *int* Tn::*int*^{A1552}. This trans-complementation does not rescue circularization of the Aux3 module in the AM-19226 background. It is also evident from this gel that there is a slight increase in circularization in the A1552 *int* mutant with trans-complementation of *int*^{A1552}, *int*^{AM-19226}, and *int*¹¹⁵⁴⁻⁷⁴. We often see a slight increase in the PCR signal in strains with the empty mTn7 (evident in AM-19226 *int* Tn in Fig. 4f). Importantly, there appears to be no real difference between the three different integrases when trans-complemented in the A1552 *int* background.

Added Line 264: To test this hypothesis, we trans-complemented the Aux3^P integrase into AM-19226 Δ *int* and found that it was unable rescue Aux3 excision, supporting the conclusion that the Aux3^P integrase has lost some functionality (Fig. 4f).

Line 267:

~~“Counter to this idea, trans-complementation of the Aux3^E integrase into A1552 Δ *int* does not appear to raise Aux3^P excision to environmental levels (Fig. 4f). This suggests that the incidence of excision is not solely reliant on integrase structure but might be differentially regulated in pandemic strains.”~~

“In the reverse experiment, trans-complementation of the Aux3^E integrase into A1552 Δ *int* does not appear to raise Aux3^P excision to environmental levels (Fig. 4f). This suggests that the incidence of excision is reliant on both integrase structure and integrase-extrinsic factors that differ between environmental and pandemic strains.”

4. It is not clear how the relative incidence fig4c was calculated. Please provide a little more description. It seems that qPCR quantification was done using a standard curve method. In that case, shouldn't it be that primer efficiency doesn't contribute to underestimation since the same primers were used for the samples and the standard curve?

Thank you for this comment on our qPCR. We have reassessed our normalization methods and found them to be a bit convoluted. We have re-done the qPCR experiment in Fig. 4c (we used a new standard curve of A1552 Aux3 gDNA for all of our targets except *attP_Aux3*) and normalized the data such that only *ompW* is used as the control for input DNA. Further, we agree with your point on primer efficiency and have removed this statement from the text. We have edited the text and the methods accordingly.

Line 223:

~~“Our results show that when normalized to incidence in DL4215 (Aux3-naïve or 100% excised), repeat 1 is present at a ratio of approximately 1 in all tested strains (Fig. 4c), indicating that repeat 1 is constant. The incidence of *attB_Aux3* when normalized to DL4215 is ~1/100 genomes for Aux3^E strains and ~1/500 genomes for Aux3^P strains (Fig. 4c), supporting *attB_Aux3* as the site of recombination. It is important to note that primer design was restricted by the proximity of the two repeats (Fig. 1b), leading to an optimal primer efficiency of ~75%. While this is below the desired efficiency, it suggests that the quantification of excision is an underestimation of the true incidence.”~~

“Our results show that when normalized to total genomic DNA, repeat 1 is present at a ratio of approximately 1 in all tested strains (Fig. 4c), indicating that repeat 1 is constant. The incidence

of *attB_Aux3* is ~1/50 genomes for Aux3^E strains and ~1/200 genomes for Aux3^P strains (Fig. 4c), supporting *attB_Aux3* as the site of recombination.”

We have further edited the methods to read as follows:

Line 496: “Primers targeted repeat 1, *attB_Aux3*, and *ompW* or *attP_Aux3* and *ompW*. All targets were measured by absolute quantification against the following standard curves: A1552 Aux3 genomic DNA (Aux3-naïve) for repeat 1, *attB_Aux3*, and *ompW* and pUC19-*attP* plasmid DNA for *attP_Aux3*. Repeat 1, *attB_Aux3*, and *attP_Aux3* signal was normalized to *ompW* to control for variability in input DNA.”

Minor

1. In Fig 4e, the *ins-is4* deletion mutant also showed excision in A1552. Could the authors discuss what other extrinsic enzymes might be responsible for this?

Thank you for this comment as we were very interested in this result. It appears that the Aux3 intrinsic recombinases are unnecessary for module excision. We believe that there is likely cross-talk going on between the Aux3 module and some other integrase encoded in the *V. cholerae* genome as this has been previously reported. Work from the Boyd lab has highlighted this cross-talk extensively. The Aux3 integrase is a P4-like, tyrosine recombinase. Other tyrosine recombinases in the pandemic *V. cholerae* genome include the integron integrase *IntI*A, *IntV*1 (VPI-1), *IntV*2 (VPI-2), *IntV*3 (VSP-II), *XerC*, and *XerD*. We feel that it is likely that another tyrosine recombinase from the *V. cholerae* genome can compensate for the reduced functionality of the pandemic Aux3 Int. The Aux3 module is directly upstream of the integron integrase *IntI*A. We deleted this gene from the *V. cholerae* genome to make *int-intI*A, *insH-intI*A, and *int-insH-intI*A mutants, and we determined that *IntI*A does not play a role in Aux3 excision (negative data not shown). We believe that while finding the compensatory integrase is very interesting, it does not add to the central focus of this paper – the reduction in Aux3 recombination to increase maintenance in the genome and enrich this module in the pandemic clade.

We have added the following text to the manuscript:

Line 244: “This could indicate the involvement of an unidentified Aux3-extrinsic recombinase in A1552, as integrase cross-talk between *V. cholerae* PAIs has been previously reported⁴⁰.”

2. Since the authors tried to address the discrepancy between *intA1552* and *intAM19226* by comparing the sequences and predicted structural models, have the authors considered an *in vitro* biochemical assay to test the enzymatic activity of *intA1552*?

Thank you for this suggestion. We have previously considered an *in vitro* recombination assay using purified Int protein. As our integrases are tyrosine recombinases, we decided it was best to perform our assays in the context of the bacterial cell. These enzymes often require other enzymes and/or host-encoded factors to catalyze excision. We believe that the presented quantitative PCR data, the new trans-complementation of *Int*^{A1552} in the AM-19226 *int* background, and our new excisionase data (Fig. 4g) effectively demonstrate that *Int*^{A1552} has reduced functionality.

3. Line 380, the gene name for TseH is VCA0285. Please correct this.

We have corrected this error in the text.

4. In Fig4d, it is not clear whether total gDNA or *ompW* was used as control to show that the increasing signal intensity is specific to increased excision but not due to overall increase in DNA quantity.

Fig. 4d was a gel image from a standard PCR experiment in which AM-19226 DNA was collected at 30 min intervals and normalized to 10 ng/uL by nanodrop. 1 uL of each normalized sample was then taken for PCR. We recognize that nanodrop quantification is not the most reliable technique, and we have therefore re-performed this experiment across a longer time course with qPCR targeting both this chromosomal *Aux3_attB* and the circular junction *Aux3_attP* (Fig. 4d). This allowed us to normalize time course samples against *ompW* as we did for the qPCR experiment in Fig. 4c.

We have removed the following line from the text:

~~“We performed PCR on AM-19226 (*Aux3^E*) DNA sampled at mid log (ML), ML+30min, ML+60min, ML+90min, and ML+120min. Band intensity increases over the time course for both excision and circularization (Fig. 4d), indicating that excision increases with progression into stationary phase.”~~

We have added the following text (Line 226):

“Time course analysis was performed to assess changes in excision and circularization in *Aux3^E* strain AM-19226 during the progression to stationary phase. Interestingly, the portion of genomes with a reformed *attB_Aux3* remains constant over the AM-19226 growth curve, while the normalized quantity of circular *Aux3^E* module increases over the AM-19226 growth curve (Fig. 4d). We find that by 4 hr of logarithmic growth there is significantly more *Aux3^E attP_Aux3* junctions than there are *attB_Aux3* junctions. This may indicate that the *Aux3^E* module carries an origin of replication, further supporting the idea that *Aux3^E* is of prophage origin.”

5. Line 79, this statement is inaccurate. TseH showed toxicity to cells only when expressed in the periplasm but not in cell-to-cell competition in the listed reference.

Thank you for making this point. Since the submission of this study, Tao Dong's group has published a follow-up study on the killing capacity of TseH in cell-to-cell competition (Hersch *et al.*, 2020). This paper demonstrates TseH is a peptidoglycan-degrading enzyme that can kill in cell-to-cell competition in a species-specific manner. We have changed the reference for this line in the manuscript to Hersch *et al.*, 2020.

6. Fig 5d. please provide the size of the DNA marker corresponding to the target band at least.

We have added arrows indicating the DNA marker sizes to figure 5d.

7. Line 216. Primer P3.2?

Thank you. We have edited the text accordingly.

8. Fig 2C. Shouldn't the Aux3-naïve att be attC? If so, indicate this in the legend.

The orange arrow in the legend of Fig 2C has been amended to indicate *attB/attL/attR*. The attachment sites have been re-named throughout the manuscript. Please see reviewer 3's comments for further explanation.

Reviewer #2 (Remarks to the Author):

This is a well-written thought provoking paper on the evolution of T6SS and their components. The authors identify T6SS genes that are associated predominantly with pandemic strains and describe the evolutionary history of the region in *V. cholerae*. The paper also describes a mechanism of how these genes became fixed in the population.

A deeper description of the phages carrying the AUX3 would be helpful. How similar are the phages, do they belong to the same viral family. Are their integrases homologous? In strains that contain AUX3 associated with a prophage are there conserved att sites.

Thank you for this suggestion. We have added a new panel to the main figures (Fig. 2c) detailing which genes are conserved across the identified prophage-like Aux3^E elements and which genes are unique to specific Aux3^E encoding strains. Shared genes were identified using Roary, which uses cd-hit with a user defined amino acid identity cut-off (70%). This figure also includes the top hit from NCBI CD search for each protein that returned one. We have also included this data in expanded format as a Supplementary Excel document (Supplementary Data 2), which includes Genbank annotations, Prokka annotations, CD search hits, and PHASTER hits for each Aux3^E element.

From the new Fig. 2c, you can see that all of the Aux3^E elements have homologous integrases. Our alignment in Supplementary Fig. 7a,b shows full conservation of the Aux3^E integrases around all of the catalytic residues and at the C-terminal tail. There is some variability in the tail. We do not show the complete alignment, but when the whole proteins are taken into consideration, the Aux3^E integrases are > 95% identical on the amino acid level. Further, each Aux3^E element is flanked by the att sites identified in Fig. 1b,c. In Fig. 1c, we highlight the att sites flanking Aux3 from AM-19226 and 1154-74 as these are the strains we use for all bench experiments throughout the paper. Every Aux3^E element we have identified is flanked by these att sites, as stated in Line 162.

We have added the following text to the manuscript:

Line 168:

“Analysis of these nine environmental strains by PHASTER⁴⁵ predicts that the Aux3 region in non-pandemic strains resembles an intact prophage of the Myoviridae family (Fig. 2b and Supplementary Fig. 5). Closer examination of the annotated coding regions in the environmental Aux3 elements reveals that the first half of each element is composed primarily of phage regulatory genes like *cro* and *cII*, toxins, methylases, holins, and other non-structural genes, but

these cassettes are variable between strains (Fig. 2c and Supplementary Data 2). The 3' half of each environmental Aux3 element is more highly conserved and is composed of tailed phage structural genes including capsid, tail, sheath, tube, and baseplate (Fig. 2c and Supplementary Data 2). To assess whether this region produces a phage particle, we collected and precipitated supernatants from *V. cholerae* 1154-74 and O395. *V. cholerae* O395 produces the filamentous CTX phage, while 1154-74 encodes a predicted Inovirus (filamentous phage) and the predicted Aux3 Myovirus (tailed phage). We were able to isolate filamentous phage from both O395 and 1154-74, but were not able to detect any tailed phage particles in the 1154-74 supernatant (Supplementary Fig. 4b). Despite its genetic resemblance to an intact prophage sequence, we cannot state that Aux3^E encodes an intact prophage.”

We have also added the following Methods section:

“Functional Prediction of Phage Genes in Aux3E Modules

Aux3^E genomic regions were extracted from *gcvT* to *thrS* and regions were re-annotated by Prokka (v1.12) (Seemann 2014). All annotated genes (from both original Genbank files and re-annotated files) were extracted and translated. Amino acid sequences for all extracted annotations were submitted to NCBI Conserved Domain Search to identify putative functional domain hits. Further functional prediction of select genes was performed by submission to HHpred (MPI Bioinformatics Toolkit).”

The location of AUX3 genes in the terminal ends of the prophage suggest they were pick up by imprecise excision. Has the circular intermediate of any of these phages been examined? Do you get AUX3 genes 1-5 forming part of the circular intermediate when the prophage excised.

Thank you for this comment. Despite constant detection by PCR, we have not isolated the actual circular intermediate DNA of the Aux3^E elements. We believe that this is likely an issue with our chosen DNA purification methods. The circular Aux3 module could likely be isolated by a large-scale plasmid preparation methods such as a CsCl preparation.

As for detecting the Aux3^P genes in the circular element, our PCR primers for detection of the circular Aux3 junction anneal within the coding region of *int* (VCA0281) and *tsiH* (VCA0286). Thus our PCR results from Fig. 4b confirm the presence of the first and sixth Aux3^P genes flanking the *attP* site.

Line 119 –Is attC2 in the AUX3 naïve strains identical to attL2 in size and sequence? A bp deletion in attC would indicate that a MGE was previously inserted there and when it excised it caused a deletion. If they are both identical, it suggests AUX3 was never present in these strains.

Thank you for this suggestion. We have extracted attC (*attB_Aux3*) from 12 environmental (Aux3-naïve) and attL2 (*attL_Aux3*) from 9 Aux3^E and 16 Aux3^P strains. We aligned all of these sequences, and we see that *attB* in Aux3-naïve strains and *attL* in Aux3^E strains are identical with the exception of 2012Env-9 and Env-390. These two Aux3^E strains are in a sister clade to the pandemic *V. cholerae* strains and are likely the most closely related Aux3^E strains to the pandemic strains. These two strains have an identical *attL* site to the pandemic strains with a 1 bp insertion in the *attL* site. Based on its distribution, we believe that this insertion likely occurred after Aux3 was integrated. As the *attL* from the majority of Aux3^E strains and *attC* from

Aux3-naïve strains are identical, we infer that all tested Aux3-naïve strains are truly naïve and are not isolates that acquired and lost Aux3.

We have not been able to identify any strains that lost Aux3, nor have we been able to show loss of Aux3 experimentally (Supplementary Fig. 6d). This is not to say that these clones do not exist, but simply that we have not been able to isolate them in our genomic/bench experiments.

Line 213 Is it possible that deletion of *int* disrupted the *att* site?

We confirmed the *int* deletion strain by Sanger sequencing using a primer that sits outside of Aux3 (upstream of *attL_Aux3*). The *attL* site is still intact in both the A1552 and AM-19226 *int* deletion strain. Further, trans-complementation of the AM-19226 *int* in the AM-19226 *int* deletion strain restores excision (Fig. 4f), supporting the presence of the *attL* site in the deletion strain.

Did you identify an excisionase associated with either MGE? Excisionases/recombination directionality factor are usually require for excision of TR integrases. In addition, another integrase elsewhere in the genome might perform the excision –cross-talk between MGE has been documented in *E. coli* and *V. cholerae* isolates. What is the integrase composition of the strains tested?

From our new Fig. 2c, we identified a gene conserved in all Aux3^E elements that has a conserved domain hit of “HTH_MerR-SF”. Both Lambda Xis and the RDF from *V. cholerae* SXT ICE have HTH MerR domains. We extracted this protein sequence and submitted it to HHpred, which returned two 98.6% probability hits for “Regulatory phage protein Cox” and “Putative excisionase”. We have summarized these hits in Supplementary Fig. 7e. We went on to overexpress this putative *xis* gene from AM-19226 in A1552, A1552 *int* Tn, A1552 *int* Tn::*int*^{A1552}, and A1552 *int* Tn::*int*^{AM-19226} backgrounds (Fig. 4g) and found that presence of this protein greatly increases the amount of excision when the more functional environmental integrase is complemented in the pandemic background. This experiment confirms this protein’s predicted function and further supports a loss of integrase function in the transition from Aux3^E to Aux3^P. We have named the identified excisionase gene *vefD*.

We have added a new sub-section to the results to discuss this finding: “**Loss of an RDF Gene Contributes to Differential Excision**” (Line 272) We have also added new text to the Introduction (Lines 65-78) and Discussion (Line 393-401).

Thank you for your comment on integrase cross-talk. We have discussed this point in Reviewer 1’s first Minor comment:

We believe that there is likely cross-talk going on between the Aux3 module and some other integrase encoded in the *V. cholerae* genome as this has been previously reported. Work from the Boyd lab has highlighted this cross-talk extensively. The Aux3 integrase is a P4-like, tyrosine recombinase. Other tyrosine recombinases in the pandemic *V. cholerae* genome include the integron integrase IntIa, IntV1 (VPI-1), IntV2 (VPI-2), IntV3 (VSP-II), XerC, and XerD. We feel that it is likely that another tyrosine recombinase from the *V. cholerae* genome can compensate for the reduced functionality of the pandemic Aux3 Int. The Aux3 module is directly upstream of the integron integrase IntIa. We deleted this gene from the *V. cholerae* genome to make *int-intIa*, *insH-intIa*, and *int-insH-intIa* mutants, and we determined that IntIa does not play a role in Aux3 excision (negative data not shown). We believe that while finding the compensatory integrase is very interesting, it does not add to the central focus of this paper

– the reduction in Aux3 recombination to increase maintenance in the genome and enrich this module in the pandemic clade.

We have added the following text to the manuscript:

Line 244: “This could indicate the involvement of an unidentified Aux3-extrinsic recombinase in A1552, as integrase cross-talk between *V. cholerae* PAIs has been previously reported⁴⁰”

Use consistent nomenclature for strain names i.e. AM199226 should be AM-19226

Thank you for this correction. All instances of AM19226 in the text and figures have been edited to AM-19226.

Line 71-72 should include Carpenter, MR et al., 2017

Thank you for the suggested reference. We have added Carpenter *et al.*, 2017 to this line.

Line 567 Ref. missing journal name

Line 583 ref. missing information

Line 587 re. missing journal title

Thank you for pointing out these errors. We have updated the references to include background for our new RDF findings, and we have ensured all references include the proper information.

Reviewer #3 (Remarks to the Author):

NCOMMS-20-02368-T ms describes the discovery that the conserved Aux3 cluster of T6SS effectors is specific of the *V. cholerae* pandemic strains and that it is carried within a mobile genetic element (MGE) remnant of a larger one, possibly a phage, circulating among a few environmental strains. This, per se, is not extremely original, as many “pathogenicity islands” carrying virulence and pathogenicity accessory functions have been described over the last 30 years in many bacterial models - including cholerae, with the CTX phage and the TCP locus. That said, this an interesting discovery, especially in the context of the role of T6SS effector sets for the killing of bacteria in complex communities. Both the comparative genomic study and the approaches chosen to identify the boundaries of the MGE, which are fairly classical, are sound and convincing.

Though if the study is well performed, the description of the identification of the recombined sequences is really confusing, and must be clarified. There are also several misuses of the terminology, both for recombination sites and gene names, which gives an impression of amateurism, that need to be corrected.

My comments below may help

Results

L108 and following

Please do not call the repeat sequences 1 and 2 attL/R1 and attL/R2, as you show later in the ms that the only real attL/R is attL/R2. This leave the impression that there are two attL/R. Call these repeats 1 and 2 and then attL/R_Aux3 for instance, once it is shown to be the integration borders and rewrite the whole paragraph accordingly.

Thank you for your suggestion. We agree that this nomenclature is confusing and have changed the text according to your suggestion.

L113 L180, ...:

attC: this terminology is already used for integron Cassette recombination sites (see the Mazel group review in the ASM mobile DNA 3). For such MGE or lysogenic phage in general, the proper term is attB (B for Bacteria by opposition to attP for Phage). And again, no need of attB1 and attB2, as there is a single one functional site according to your assay.

Thank you for this suggestion. We have changed all instances of *attC1* to “repeat 1” and all instances of *attC2* to “*attB_Aux3*”.

L111

about the stretch of T, please delete this useless part. HK022 which is highly related to Lambda (much more than P4) has an overlap region with more G, ie it has no meaning and give the impression that it is somewhat related to lambda...

Thank you for making this point. We have removed this line/reference from the manuscript.

Fig 1 and text. Please correct the gene names: *is5* is not a gene! So, use *insH*, which is the name of the IS5 transposase (or VCA0282).

Thank you for making this correction. We have changed all instances of *is5* in the figures and text to “*insH*”.

L269:

This is not a conjugation frequency, but a cointegrate formation frequency, please correct.

Thank you for this suggestion. We have changed each instance of transconjugants to “cointegrates” and each instance of conjugation frequency to “cointegrate formation frequency” in the figures and text.

Discussion

L280.

As you have no proof that the MGE is a phage, even in the environmental form (as mentioned L300), just delete phage before integrate.

We have removed the word “phage” from line 333.

L303 and 304. Transformation goes through ssDNA, exclusively, there is de-gradation of one of

the two DNA strand, so apart if they produce circular concatainers, the chance to transfer the circular form are null.

Thank you for this comment. We agree with this statement and have modified the text for accuracy.

Line 356:

~~“Two potential mechanisms by which Aux3^E could be transferred between strains without making its own phage particle are generalized transduction by environmental lytic phages or chitin-induced natural competence. For the latter mechanism, excision and circularization would likely confer no advantage, as the *V. cholerae* natural competence machinery can foster the transfer of linear genome fragments significantly larger than the Aux3^E module.”~~

“Two potential mechanisms by which Aux3^E could be transferred between strains without making its own phage particle are generalized transduction by environmental lytic phages or chitin-induced natural competence. For the latter mechanism, Aux3^E would be transferred as a linear fragment of genomic DNA and incorporated by homologous recombination in the flanking regions outside of the *attL* and *attR* sites. The *V. cholerae* natural competence machinery can foster the acquisition of linear genome fragments significantly larger than the Aux3^E module⁵².”

L331 and following.

You do not discuss the possible presence of a recombination directionality factor encoding gene (ie a *xis* gene) in the Aux3E locus, which if absent in the P version could explain your results? IntE would not be sufficient for excision in P strain, but sufficient for integration, ie similarly to what you observed. This is worth discussing, and furthermore, more generally you should discuss the genes/ORF found in the Aux3E MGE, what are the chance that it is really a lysogenic phage?

We have added a new panel to the main figures (Fig. 2c) detailing which genes are conserved across the identified prophage-like Aux3^E elements and which genes are unique to specific Aux3^E encoding strains. Shared genes were identified using Roary, which uses cd-hit with a user defined amino acid identity cut-off (70%). This figure also includes the top hit from NCBI CD search for each protein that returned one. We have also included this data in expanded format as a Supplementary Excel document (Supplementary Data 2), which includes Genbank annotations, Prokka annotations, CD search hits, and PHASTER hits for each Aux3^E element. From this new figure/table we identified a gene conserved in all Aux3^E elements that has a conserved domain hit of “HTH_MerR-SF”. Both Lambda Xis and the RDF from *V. cholerae* SXT ICE have HTH MerR domains. We extracted this protein sequence and submitted it to HHpred, which returned two 98.6% probability hits for “Regulatory phage protein Cox” and “Putative excisionase”. We have summarized these hits in Supplementary Fig. 7e. We went on to overexpress this putative *xis* gene from AM-19226 in A1552, A1552 *int* Tn, A1552 *int* Tn::*int*^{A1552}, and A1552 *int* Tn::*int*^{AM-19226} backgrounds (Fig. 4g) and found that presence of this protein greatly increases the amount of excision when the more functional environmental integrase is complemented in the pandemic background. This experiment confirms this protein’s predicted function and further supports a loss of integrase function in the transition from Aux3^E to Aux3^P. We have named the identified excisionase gene *vefD*.

We have added a new sub-section to the results to discuss this finding: “**Loss of an RDF Gene Contributes to Differential Excision**” (Line 272) We have also added new text to the Introduction (Lines 65-78) and Discussion (Line 393-401).

To assess the presence of an Aux3 phage particle, we grew *V. cholerae* strains 1154-74 (Aux3^E) and O395 (constitutive CTX phage producing strain) with the same conditions under which we see strong Aux3 excision and circularization (4 hr culture in LB at 37 C shaking). We then collected supernatant and performed PEG8000 aggregation to concentrate any present phage particles. We analyzed these phage preparations by electron microscopy. 1154-74 encodes two predicted prophages: the Aux3 prophage which is most closely related to known members of the Myoviridae family (dsDNA tailed phages) and a predicted inovirus (filamentous phage). CTX phage (inovirus) was present in O395 phage preps. A filamentous phage was also seen in 1154-74 preps, but no tailed phages were found. We have included images of these filamentous phages in Supplementary Fig. 4b. We have edited the text as follows:

Line 170:

“Closer examination of the annotated coding regions in the environmental Aux3 elements reveals that the 5' half of each element is composed primarily of phage regulatory genes like *cro* and *cII*, toxins, methylases, holins, and other non-structural genes, but these cassettes are variable between strains (Fig. 2c and Supplementary Data 2). The 3' half of each environmental Aux3 element is more highly conserved and is composed of tailed phage structural genes including capsid, tail, sheath, tube, and baseplate (Fig. 2c and Supplementary Data 2). To assess whether this region produces a phage particle, we collected and precipitated supernatants from *V. cholerae* 1154-74 and O395. *V. cholerae* O395 produces the filamentous CTX phage, while 1154-74 encodes a predicted Inovirus (filamentous phage) and the predicted Aux3 Myovirus (tailed phage). We were able to isolate filamentous phage from both O395 and 1154-74, but were not able to detect any tailed phage particles in the 1154-74 supernatant (Supplementary Fig. 4b). Despite its genetic resemblance to an intact prophage sequence, we cannot state that Aux3^E encodes an intact prophage.”

Reference list

Ref 13: “Proc....” ?

Ref 22: the journal (Science) is missing

Ref 28 : the journal (Mol Micro) is missing

Ref 44: the journal (PNAS) is missing

Thank you for pointing out these errors. We have updated the references to include background for our new RDF findings, and we have ensured all references include the proper information.

REVIEWERS' COMMENTS

Reviewer #1 (Remarks to the Author):

The authors have fully addressed all my concerns.

Reviewer #2 (Remarks to the Author):

The authors have addressed the reviewers' comments in full and this reviewer has no further corrections or comments on the manuscript.

The manuscript makes a significant contribution to the field and will be of broad interest to those interested in the evolution of pathogens.